# CONTRACTIVE SYSTEMS IMPROVE GRAPH NEURAL NETWORKS AGAINST ADVERSARIAL ATTACKS

## ABSTRACT

Graph Neural Networks (GNNs) have established themselves as a key component in addressing diverse graph-based tasks. Despite their notable successes, GNNs remain susceptible to input perturbations in the form of adversarial attacks. This paper introduces an innovative approach to fortify GNNs against adversarial perturbations through the lens of contractive dynamical systems. Our method introduces graph neural layers based on differential equations with contractive properties, which, as we show, improve the robustness of GNNs. A distinctive feature of the proposed approach is the simultaneous learned evolution of both the node features and the adjacency matrix, yielding an intrinsic enhancement of model robustness to perturbations in the input features and the connectivity of the graph. We mathematically derive the underpinnings of our novel architecture and provide theoretical insights to reason about its expected behavior. We demonstrate the efficacy of our method through numerous real-world benchmarks, reading on par or improved performance compared to existing methods.

## 1 INTRODUCTION

In recent years, the emergence of Graph Neural Networks (GNNs) has revolutionized the field of graph machine learning, offering remarkable capabilities for modeling and analyzing complex graph-structured data. These networks have found applications in diverse domains and applications, from Network Analysis (Defferrard et al., 2016; Kipf & Welling, 2017) and recommendation systems, Bioinformatics (Senior et al., 2020; Corso et al., 2023), Computer Vision (Wang et al., 2018), and more. However, the increasing prevalence of GNNs in critical decision-making processes has also exposed them to new challenges, particularly in terms of vulnerability to adversarial attacks.

In particular, it has been shown that one can design small adversarial perturbations of the input graph and its node features, that result in vastly different GNN predictions (Waniek et al., 2018; Zügner & Günnemann, 2019). Adversarial attacks received extensive attention in the context of Convolutional Neural Networks (CNNs) (Goodfellow et al., 2015), but graph data has an added degree of freedom compared to data on a regular grid: the connectivity of the graph can be altered by adding or removing edges. Also, in natural settings, such as social network graphs, connectivity perturbations may be more realistically implementable by a potential adversary, rather than perturbations of the node features. This gives rise to hard discrete optimization problems, which necessitates the study of adversarial robustness specialized to graph data and GNNs (Günnemann, 2022).

In this paper we propose a GNN architecture that jointly processes the adversarially attacked adjacency matrix and node features by a learnable neural dynamical system. Our approach extends the active research front that aims to design neural architectures that enjoy inherent properties and behavior, drawing inspiration from dynamical systems with similar properties (Haber & Ruthotto, 2017; Weinan, 2017; Chen et al., 2018; Choromanski et al., 2020; Chamberlain et al., 2021; Thorpe et al., 2022; Eliasof et al., 2021; Celledoni et al., 2023). This approach has also been used to improve the robustness of CNNs, see Appendix E. Specifically, the flow map of the dynamical system under consideration in this work draws inspiration from the theory of non-Euclidean contractive systems developed in Bullo (2023) to offer an adversarially robust GNN. We therefore name our method CSGNN. Notably, because adjacency matrices are not arbitrary matrices, their learnable neural dynamical system needs to be carefully crafted to ensure it is node-permutation equivariant, and

that it yields a symmetric adjacency matrix. To the best of our knowledge, this is the first attempt at learning coupled dynamical systems that evolve both the node features and the adjacency matrix.

**Main contributions.** This paper offers the following advances in adversarial defense against poisoning attacks in GNNs: (i) A novel architecture, CSGNN, that jointly evolves the node features and the adjacency matrix to improve GNN robustness to input perturbations, (ii) A theoretical analysis of our CSGNN, addressing the relevance of the architecture based on the theory of contractive dynamical systems , and, (iii) Improved performance on various graph adversarial defense benchmarks.

## 2 RELATED WORK

**Graph Neural Networks as Dynamical Systems.**  Drawing inspiration from dynamical systems models that admit desired behaviors, various GNN architectures have been proposed to take advantage of such characteristic properties. In particular, building on the interpretation of CNNs as discretizations of PDEs (Chen et al., 2018; Ruthotto & Haber, 2020), there have been multiple works that view GNN layers as time integration steps of learnable non-linear diffusion equations. Such an approach allowed exploiting this connection to understand and improve GNNs (Chamberlain et al., 2021; Eliasof et al., 2021), for example by including energy-conserving terms alongside diffusion (Eliasof et al., 2021; Rusch et al., 2022) or using reaction-diffusion equations (Wang et al., 2022; Choi et al., 2022). These approaches to designing GNNs have been shown to be of significant benefit when trying to overcome common issues such as over-smoothing (Nt & Maehara, 2019; Oono & Suzuki, 2020; Cai & Wang, 2020; Rusch et al., 2023) and over-squashing (Alon & Yahav, 2021). Recently, it was shown that neural diffusion GNNs are robust to graph attacks (Song et al., 2022).

We note that deep learning architectures are also often harnessed to numerically solve ODEs and PDEs or discover such dynamical systems from data (Long et al., 2018; Bar-Sinai et al., 2019; Brandstetter et al., 2022; Cai et al., 2022). However, in this paper, we focus on drawing links between GNNs and contractive dynamical systems to improve GNN robustness to adversarial attacks.

**Adversarial Defense in Graph Neural Networks.**  Various adversarial attack algorithms have been designed for graph data, notably including nettack (Zügner et al., 2018), which makes local changes to targeted nodes' features and connectivity, and metattack (Zügner & Günnemann, 2019), which uses a meta-learning approach with a surrogate model, usually a graph convolutional network (GCN) (Kipf & Welling, 2017), to generate a non-targeted global graph attack, and, recently Chen et al. (2022) proposed a novel method to create graph injection attacks.

In response to these developments, significant efforts were made to design methods that improve GNN robustness. The majority of these approaches focus on perturbations of the graph connectivity, as those are more likely and practical in social network graph datasets. Several of these methods preprocess the graph based on underlying assumptions or heuristics, for example, dropping edges where node features are not similar enough, under the assumption that the true, non-attacked, graph is homophilic (Wu et al., 2019). Another approach, in (Entezari et al., 2020), suggests truncating the singular value decomposition of the adjacency matrix, effectively eliminating its high-frequency components, based on the assumption that adversarial attacks add high-frequency perturbations to the true adjacency matrix. The aforementioned approaches are unsupervised, and are typically added to existing GNN architectures while training them for a specific downstream task, such as node classification. Additionally, there are defenses that clean the attacked graph in a supervised manner, such as Pro-GNN (Jin et al., 2020), which solves a joint optimization problem for the GNN's learnable parameters, as well as for the adjacency matrix, with sparsity and low-rank regularization.

Besides methods for cleaning attacked adjacency matrices, there are also methods that aim to design robust GNN *architectures*. An example of this is given in (Huang et al., 2023), where the GCN architecture is modified to use a mid-pass filter, resulting in increased robustness. In this context, it is also natural to consider the use of Lipschitz constraints: given an upper bound on the Lipschitz constant of a classifier and a lower bound on its margin, we can issue robustness certificates (Tsuzuku et al., 2018b). This has been studied to some extent in the context of GNNs (Jia et al., 2023), although, in this case, and in contrast to our work, the Lipschitz continuity is studied only with respect to the node features. In our work, we also consider the Lipschitz continuity with respect to the adjacency matrix. It is worth noting that the development of a defense mechanism should ideally be done in tandem with the development of an adaptive attack, although designing an appropriate adaptive

attack is not generally a straightforward task (Mujkanovic et al., 2022). In Mujkanovic et al. (2022), a set of attacked graphs are provided as "unit tests", which have been generated using adaptive attacks for various defenses. Therefore, in our experiments, we consider both standard, long-standing benchmarks, as well as recently proposed attacks in (Mujkanovic et al., 2022).

## 3 PRELIMINARIES

**Notations.** Let $G = (V, E)$ be a graph with $n$ nodes $V$ and $m$ edges $E$, also associated with the adjacency matrix $\mathbf{A} \in \mathbb{R}^{n \times n}$, such that $\mathbf{A}_{i,j} = 1$ if $(i,j) \in E$ and 0 otherwise, and let $\mathbf{f}_i \in \mathbb{R}^{c_{\text{in}}}$ be the input feature vector of the node $\mathbf{v}_i \in V$. In this paper, we focus on *poisoning* attacks, and we assume two types of possible attacks (perturbations) of the clean, true data, before training the GNN: (i) The features $\mathbf{f}_i$ are perturbed to $(\mathbf{f}_*)_i$, and, (ii) the adjacency matrix $\mathbf{A}$ of the graph is perturbed by adding or removing edges, denoted by $E_*$, inducing a perturbed adjacency matrix $\mathbf{A}_* \in \mathbb{R}^{n \times n}$. We denote by $G_* = (V_*, E_*)$ the attacked graph with the same vertices, i.e. $V = V_*$, by $\mathbf{A}_* \in \mathbb{R}^{n \times n}$ the perturbed adjacency matrix, and the perturbed node features are denoted by $(\mathbf{f}_*)_i$. We also denote by $\mathbf{F}, \mathbf{F}_* \in \mathbb{R}^{n \times c_{\text{in}}}$ the matrices collecting, as rows, the individual node features $\mathbf{f}_i, (\mathbf{f}_*)_i$.

**Measuring graph attacks.** To quantify the robustness of a GNN with respect to an adversarial attack, it is necessary to measure the impact of the attack. For node features, it is common to consider the Frobenius norm $\| \cdot \|_F$ to quantify the difference between the perturbed features $\mathbf{F}_*$ from the clean ones $\mathbf{F}$. However, the Frobenius norm is not a natural metric for adjacency attacks, see (Jin et al., 2021; Bojchevski & Günnemann, 2019; Günnemann, 2022) for example. Instead, it is common to measure the $\ell^0$ distance between the true and attacked adjacency matrices, as follows:

$$\ell^0(\mathbf{A}, \mathbf{A}_*) = |\mathcal{I}(\mathbf{A}, \mathbf{A}_*)|, \tag{1}$$

where $\mathcal{I}(\mathbf{A}, \mathbf{A}_*) = \{i, j \in \{1, \ldots, n\} : \mathbf{A}_{ij} \neq (\mathbf{A}_*)_{ij}\}$. For brevity, we refer to $\mathcal{I}(\mathbf{A}, \mathbf{A}_*)$ as $\mathcal{I}$, and by $|\mathcal{I}|$ we refer to the cardinality of $\mathcal{I}(\mathbf{A}, \mathbf{A}_*)$, as in Equation (1). The $\ell^0$ distance measures how many entries of $\mathbf{A}$ need to be modified to obtain $\mathbf{A}_*$, and is typically used to measure budget constraints in studies of adversarial robustness of GNNs (Günnemann, 2022; Mujkanovic et al., 2022). Given that typical adjacency matrices consist of binary entries, it follows that:

$$\ell^0(\mathbf{A}, \mathbf{A}_*) = \|\text{vec}(\mathbf{A}) - \text{vec}(\mathbf{A}_*)\|_1 = \sum_{i,j=1}^{n} |\mathbf{A}_{ij} - (\mathbf{A}_*)_{ij}| = \ell^1(\mathbf{A}, \mathbf{A}_*), \tag{2}$$

where $\text{vec}(\cdot)$ is the flattening operator, obtained by stacking the columns of $\mathbf{A}$. We refer to $\|\text{vec}(\mathbf{A}) - \text{vec}(\mathbf{A}_*)\|_1$ as the vectorized $\ell^1$ norm. That is, for binary matrices, the $\ell^0$ and $\ell^1$ norms coincide. However, using the $\ell^0$ distance to implement constraints or regularization gives rise to computationally hard optimization problems because it is non-convex and non-smooth (Wright & Ma, 2022), and unfortunately, the equality in Equation (2) is generally not true for arbitrary real-valued matrices. Fortunately, as shown in Wright & Ma (2022), for matrices with $\|\text{vec}(\mathbf{A})\|_\infty \leq 1$, the vectorized $\ell^1$ norm is the largest convex function bounded from above by the $\ell^0$ norm, that is:

$$\|\text{vec}(\mathbf{A})\|_1 \leq \|\text{vec}(\mathbf{A})\|_0 = |\{i, j \in \{1, \ldots n\} : \mathbf{A}_{ij} \neq 0\}|. \tag{3}$$

This property makes the usage of the $\ell^1$ norm a common approximation of the $\ell^0$ norm. Furthermore, it is also possible to relate the two norms, as follows:

$$\|\text{vec}(\mathbf{A}) - \text{vec}(\mathbf{A}_*)\|_1 = \sum_{(i,j) \in \mathcal{I}} |\mathbf{A}_{ij} - (\mathbf{A}_*)_{ij}| \geq |\mathcal{I}| \cdot \min_{(i,j) \in \mathcal{I}} |\mathbf{A}_{ij} - (\mathbf{A}_*)_{ij}|, \tag{4}$$

i.e. the $\ell^1$ norm can be lower bounded by the $\ell^0$ norm, up to a multiplicative constant. Therefore, we can still use the $\ell^1$ norm to measure the distance between arbitrary matrices as an approximation of the $\ell^0$ norm. Throughout this paper, we denote the perturbed node features by $\mathbf{F}_* = \mathbf{F} + \delta\mathbf{F}$, and the perturbed adjacency matrix by $\mathbf{A}_* = \mathbf{A} + \delta\mathbf{A}$, where $\|\delta\mathbf{F}\|_F < \varepsilon_1$, and $\|\text{vec}(\delta\mathbf{A})\|_1 \leq \varepsilon_2$.

In adversarial defense, the goal is to design a mechanism, such that the output of the neural network is stable with respect to the perturbations $\delta\mathbf{F}$ and $\delta\mathbf{A}$. As discussed in Section 2, this goal is typically met either by modified architectures, training schemes, as well as their combinations. In Section 4, we present CSGNN - a defense mechanism based on a dynamical system perspective. This approach aims to reduce the sensitivity to input perturbations of the neural network and is based on the theory

of contractive dynamical systems (Bullo, 2023). We will refer to a map as contractive with respect to a norm $\| \cdot \|$ if it is $1-$Lipschitz in such norm. Furthermore, we define contractive dynamical systems as those whose solution map is contractive with respect to the initial conditions. For completeness, in Appendix A we mathematically define and discuss contractive systems.

We start from the assumption that the best training accuracy on a given task corresponds to the clean inputs $(\mathbf{A}, \mathbf{F})$. The main idea of CSGNN is to jointly evolve the features $\mathbf{F}_*$ and the adjacency matrix $\mathbf{A}_*$, so that even if their clean versions $\mathbf{F}$ and $\mathbf{A}$ are not known, the network would output a vector measurably similar to the one corresponding to $(\mathbf{A}, \mathbf{F})$, as we formulate in the following section.

## 4  METHOD

### 4.1  GRAPH NEURAL NETWORKS INSPIRED BY CONTRACTIVE SYSTEMS

We now present our CSGNN, focused on the task of robust node classification, where we wish to predict the class of each node in the graph, given attacked input data $(\mathbf{F}_*, \mathbf{A}_*)$. The goal is therefore to design and learn a map $\mathcal{D} : \mathbb{R}^{n \times c} \times \mathbb{R}^{n \times n} \rightarrow \mathbb{R}^{n \times c} \times \mathbb{R}^{n \times n}$, that evolves the node features, as well as the adjacency matrix. To the best of our knowledge, this is the first attempt at learning a *coupled dynamical system* that considers both the node features and the adjacency matrix.

We implement the map $\mathcal{D}$ as a composition of learnable dynamical systems inspired by contractivity theory, that simultaneously update $\mathbf{F}_*$ and $\mathbf{A}_*$. Specifically, we model $\mathcal{D}$ as an approximation of the solution, at the final time $T$, of the continuous dynamical system:

$$\begin{cases} \dot{F}(t) = X(t, F(t), A(t)) \in \mathbb{R}^{n \times c} \\ \dot{A}(t) = Y(t, A(t)) \in \mathbb{R}^{n \times n}, \\ (F(0), A(0)) = (\mathcal{K}(\mathbf{F}_*), \mathbf{A}_*), \end{cases} \tag{5}$$

where $\dot{F} = \mathrm{d}F/\mathrm{d}t$ denotes the first order derivative in time, and $\mathcal{K} : \mathbb{R}^{c_{\text{in}}} \rightarrow \mathbb{R}^{c}$ is a linear embedding layer. Similarly to (Haber & Ruthotto, 2017; Eliasof et al., 2021; Chamberlain et al., 2021), we assume both $X$ and $Y$ to be piecewise constant in time, i.e., that on a given time interval $[0, T]$, there is a partition $0 = \tau_0 < \tau_1 < \ldots < \tau_L = T$, $h_l = \tau_l - \tau_{l-1}$ for $l = 1, \ldots, L$, such that:

$$X(t, \mathbf{F}, \mathbf{A}) = X_l(\mathbf{F}, \mathbf{A}), \quad Y(t, \mathbf{A}) = Y_l(\mathbf{A}), \quad \mathbf{F} \in \mathbb{R}^{n \times c}, \mathbf{A} \in \mathbb{R}^{n \times n}, t \in [\tau_{l-1}, \tau_l), \tag{6}$$

for a pair of functions $X_l : \mathbb{R}^{n \times c} \times \mathbb{R}^{n \times n} \rightarrow \mathbb{R}^{n \times c}$, $Y_l : \mathbb{R}^{n \times n} \rightarrow \mathbb{R}^{n \times n}$. When referring to the approximation of $(F(\tau_l), A(\tau_l))$, we use $(\mathbf{F}^{(l)}, \mathbf{A}^{(l)})$ when we start with the clean pair, and $(\mathbf{F}_*^{(l)}, \mathbf{A}_*^{(l)})$ with the perturbed one. To obtain a neural network, we consider the solution of Equation (5) at time $T$, which is approximated using the explicit Euler method. More explicitly, we compose $L$ explicit Euler layers each defined as $D_l((\mathbf{F}, \mathbf{A})) := (\Psi_{X_l}^{h_l}(\mathbf{F}, \mathbf{A}), \Psi_{Y_l}^{h_l}(\mathbf{A}))$, $l = 1, \ldots, L$, where $\Psi_{X_l}^{h_l}(\mathbf{F}, \mathbf{A}) := \mathbf{F} + h_l X_l(\mathbf{F}, \mathbf{A})$, and $\Psi_{Y_l}^{h_l}(\mathbf{A}) := \mathbf{A} + h_l Y_l(\mathbf{A})$ are the explicit Euler steps for $X_l$ and $Y_l$, respectively. The map $\mathcal{D}$ is then defined as the composition of $L$ layers:

$$\mathcal{D} := D_L \circ \ldots \circ D_1. \tag{7}$$

The coupled dynamical system encapsulated in $\mathcal{D}$ evolves both the hidden node features and the adjacency matrix for $L$ layers. We denote the output of $\mathcal{D}$ by $(\mathbf{F}_*^{(L)}, \mathbf{A}_*^{(L)}) = \mathcal{D}((\mathcal{K}(\mathbf{F}_*), \mathbf{A}_*))$. To obtain node-wise predictions from the network to solve the downstream task, we feed the final GNN node features $\mathbf{F}_*^{(L)}$ to a classifier $\mathcal{P} : \mathbb{R}^{c} \rightarrow \mathbb{R}^{c_{\text{out}}}$, which is implemented by a linear layer.

To better explain the structure of CSGNN, we provide an illustration in Figure 1 and a detailed feed-forward description in Appendix G.

In what follows, we describe how to characterize the functions $\Psi_{X_l}^{h_l}$ and $\Psi_{Y_l}^{h_l}$ from Equation (7). First, in Section 4.2, we derive the node feature dynamical system governed by $X$. We show, that under mild conditions, contractivity can be achieved, allowing us to derive a bound on the influence of the attacked node features $\mathbf{F}_*$ on the GNN output. Second, in Section 4.3, we develop and propose a novel contractive dynamical system for the adjacency matrix, which is guided by $Y$.

Our motivation in designing such a coupled system stems from the nature of our considered adversarial settings. That is, we assume, that the adjacency matrix is perturbed. We note, that the adjacency

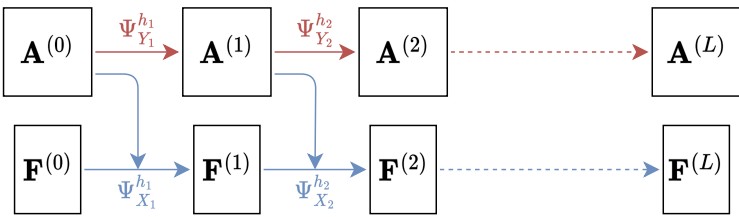

Figure 1: The coupled dynamical system $\mathcal{D}$ in CSGNN, as formulated in Equation (7).

matrix controls the propagation of node features, and therefore, leaving the input attacked adjacency unchanged, may result in sub-par performance. While some methods employ a pre-processing step of the attacked matrix $\mathbf{A}_*$ (Entezari et al., 2020; Wu et al., 2019), it has been shown that joint optimization of the node features and the adjacency matrix can lead to improved performance (Jin et al., 2020). Therefore, we will develop and study novel, coupled dynamical systems that evolve both the node features and the adjacency matrix and are learned in a data-driven manner. This perspective allows to obtain favorable properties such as adjacency matrix contractivity, thereby reducing the sensitivity to adversarial adjacency matrix attacks.

## 4.2 CONTRACTIVE NODE FEATURE DYNAMICAL SYSTEM

We now describe the learnable functions $\Psi_{X_l}^{h_l}$, $l = 1, \ldots, L$, that determine the node feature dynamics of our CSGNN. We build upon a diffusion-based GNN layer, (Chamberlain et al., 2021; Eliasof et al., 2021), that is known to be stable, and under certain assumptions is contractive. More explicitly, our proposed $\Psi_{X_l}^{h_l}$ is characterized as follows:

$$
\begin{aligned}
\Psi_{X_l}^{h_l}(\mathbf{F}^{(l-1)}, \mathbf{A}^{(l-1)}) := \mathbf{F}^{(l)} &= \mathbf{F}^{(l-1)} + h_l X_l(\mathbf{F}^{(l-1)}, \mathbf{A}^{(l-1)}) \\
&= \mathbf{F}^{(l-1)} - h_l \left( \mathcal{G}(\mathbf{A}^{(l-1)})^{\top} \sigma \left( \mathcal{G}(\mathbf{A}^{(l-1)}) \mathbf{F}^{(l-1)} \mathbf{W}_l \right) \mathbf{W}_l^{\top} \right) \tilde{\mathbf{K}}_l,
\end{aligned} \tag{8}
$$

where $\mathbf{W}_l \in \mathbb{R}^{c \times c}$, and $\tilde{\mathbf{K}}_l = (\mathbf{K}_l + \mathbf{K}_l^{\top})/2 \in \mathbb{R}^{c \times c}$ are learnable parameters which allows a gradient flow interpretation of our system, as in Giovanni et al. (2023). Also, as in Eliasof et al. (2021), the map $\mathcal{G}(\mathbf{A}^{(l-1)}) : \mathbb{R}^{n \times n} \to \mathbb{R}^{n \times n}$ is the gradient operator of $\mathbf{A}^{(l-1)}$ defined in Appendix B, and we set $\sigma = \text{LeakyReLU}$.

**Theorem 1** (Equation (8) can induce stable node dynamics). *Assume $\sigma$ is a monotonically increasing 1-Lipschitz non-linear function. There are choices of $(\mathbf{W}_l, \mathbf{K}_l) \in \mathbb{R}^{n \times n} \times \mathbb{R}^{c \times c}$, for which the explicit Euler step in Equation (8) is stable for a small enough $h_l > 0$, i.e. there is a convex energy $\mathcal{E}_{\mathbf{A}}$ for which*

$$
\mathcal{E}_{\mathbf{A}}(\Psi_{X_l}^{h_l}(\mathbf{F}^{(l-1)}, \mathbf{A})) \leq \mathcal{E}_{\mathbf{A}}(\mathbf{F}^{(l-1)}), \quad l = 1, \ldots, L. \tag{9}
$$

**Theorem 2** (Equation (8) can induce contractive node dynamics). *Assume $\sigma$ is a monotonically increasing 1-Lipschitz non-linear function. There are choices of $(\mathbf{W}_l, \mathbf{K}_l) \in \mathbb{R}^{n \times n} \times \mathbb{R}^{c \times c}$, for which the explicit Euler step in Equation (8) is contractive for a small enough $h_l > 0$, i.e.*

$$
\|\Psi_{X_l}^{h_l}(\mathbf{F} + \delta\mathbf{F}, \mathbf{A}) - \Psi_{X_l}^{h_l}(\mathbf{F}, \mathbf{A})\|_F \leq \|\delta\mathbf{F}\|_F, \quad \delta\mathbf{F} \in \mathbb{R}^{n \times c}. \tag{10}
$$

In Appendix B we prove Theorems 1 and 2 for various parameterizations. One parametrization for which both theorems are satisfied corresponds to $\mathbf{K}_l = I_c$. We have experimented with this configuration, learning only $\mathbf{W}_l \in \mathbb{R}^{n \times n}$. We found that this configuration improves several baseline results, showing the benefit of contractive node feature dynamics. We report those results in Appendix J. We also found, following recent interpretations of dissipative and expanding GNNs Giovanni et al. (2023), that choosing the parametrization as $\mathbf{W}_l = I_n$, and training $\mathbf{K}_l \in \mathbb{R}^{c \times c}$ leads to further improved results, as we show in our experiments in Section 5 and Appendix J. We note that this parameterization admits stable dynamical systems, in the sense of Theorem 1, as discussed in Appendix B.

## 4.3 CONTRACTIVE ADJACENCY MATRIX DYNAMICAL SYSTEM

As previously discussed, our CSGNN learns both node features and adjacency matrix dynamical systems to defend against adversarial attacks. We now elaborate on the latter, aiming to design and

learn dynamical systems $\Psi_{Y_l}^{h_l} : \mathbb{R}^{n \times n} \to \mathbb{R}^{n \times n}$, $l = 1, \ldots, L$, such that:

$$\|\text{vec}(\Psi_{Y_l}^{h_l}(\mathbf{A}^{(l-1)})) - \text{vec}(\Psi_{Y_l}^{h_l}(\mathbf{A}_*^{(l-1)}))\|_1 \leq \|\text{vec}(\mathbf{A}^{(l-1)}) - \text{vec}(\mathbf{A}_*^{(l-1)})\|_1, \qquad (11)$$

where

$$\Psi_{Y_l}^{h_l}(\mathbf{A}^{(l-1)}) = \mathbf{A}^{(l)} = \mathbf{A}^{(l-1)} + h_l Y_l(\mathbf{A}^{(l-1)}). \qquad (12)$$

In other words, we wish to learn maps $Y_l$ that *decrease* the vectorized $\ell^1$ distance between the true and attacked adjacency matrices, thereby reducing the effect of the adjacency matrix attack.

Since we are concerned with adjacency matrices, we need to pay attention to the structure of the designed map $Y_l$. Specifically, we demand that (i) the learned map $Y_l$ are *node-permutation-equivariant*. That is, relabelling (change of order) of the graph nodes should not influence the dynamical system $\Psi_{Y_l}^{h_l}$ output up to its order, and, (ii) if the input graph is symmetric, then the updated adjacency matrix $\mathbf{A}^{(l)}$ should remain symmetric. Formally, requirement (i) demands that:

$$\Psi_{Y_l}^{h_l}(\mathbf{P}\mathbf{A}\mathbf{P}^\top) = \mathbf{P}\Psi_{Y_l}^{h_l}(\mathbf{A})\mathbf{P}^\top \qquad (13)$$

should hold for every permutation matrix $\mathbf{P} \in \{0,1\}^{n \times n}$. The symmetry condition (ii) implies that we want $(\Psi_{Y_l}^{h_l}(\mathbf{A}))^\top = \Psi_{Y_l}^{h_l}(\mathbf{A})$. To this end, we adopt the derivations provided in Maron et al. (2018, Appendix A), that show that in order to make the map $\Psi_{Y_l}^{h_l}$ permutation-equivariant and also symmetry preserving, we can set $Y_l(\mathbf{A}) = \sigma(M(\mathbf{A}))$ in Equation (12), where $\sigma : \mathbb{R} \to \mathbb{R}$ is any non-linear activation function, and $M : \mathbb{R}^{n \times n} \to \mathbb{R}^{n \times n}$ is a linear map defined as follows:

$$
\begin{aligned}
M(\mathbf{A}) = {} & k_1 \mathbf{A} + k_2 \text{diag}(\text{diag}(\mathbf{A})) + \frac{k_3}{2n}(\mathbf{A}\mathbf{1}_n\mathbf{1}_n^\top + \mathbf{1}_n\mathbf{1}_n^\top\mathbf{A}) + k_4\text{diag}(\mathbf{A}\mathbf{1}_n) \\
& + \frac{k_5}{n^2}(\mathbf{1}_n^\top\mathbf{A}\mathbf{1}_n)\mathbf{1}_n\mathbf{1}_n^\top + \frac{k_6}{n}(\mathbf{1}_n^\top\mathbf{A}\mathbf{1}_n)I_n + \frac{k_7}{n^2}(\mathbf{1}_n^\top\text{diag}(\mathbf{A}))\mathbf{1}_n\mathbf{1}_n^\top \qquad (14) \\
& + \frac{k_8}{n}(\mathbf{1}_n^\top\text{diag}(\mathbf{A}))I_n + \frac{k_9}{2n}(\text{diag}(\mathbf{A})\mathbf{1}_n^\top + \mathbf{1}_n(\text{diag}(\mathbf{A}))^\top),^1
\end{aligned}
$$

for an arbitrary, learnable vector $\mathbf{k} = (k_1, \cdots, k_9) \in \mathbb{R}^9$. We now provide a theorem that validates the contractivity of the proposed adjacency matrix dynamical system, with its proof in Appendix C.

**Theorem 3** (Equation (14) defines contractive adjacency dynamics). *Let $\alpha \leq 0$, $\sigma : \mathbb{R} \to \mathbb{R}$ be a Lipschitz continuous function, with $\sigma'(s) \in [0,1]$ almost everywhere. If $0 \leq h_l \leq \hat{h}_l^{\text{adj}} := \frac{2}{(2\sum_{i=2}^9 |k_i|) - \alpha}$, then the explicit Euler step*

$$\mathbf{A}^{(l)} = \Psi_{Y_l}^{h_l}(\mathbf{A}^{(l-1)}) := \mathbf{A}^{(l-1)} + h_l\sigma\left(M(\mathbf{A}^{(l-1)})\right), \qquad (15)$$

*where $k_1 = \left(\alpha - \sum_{i=2}^9 |k_i|\right)$, is contractive in the vectorized $\ell^1$ norm.*

In our experiments, $\alpha \leq 0$ is a non-positive hyperparameter of the network.

## 5 EXPERIMENTS

We now study the effectiveness of CSGNN against different graph adversarial attacks. In Section 5.1 we discuss our experimental settings. In Section 5.2, we report our experimental results and observations on several benchmarks, with additional results and an ablation study in Appendix J.

### 5.1 EXPERIMENTAL SETTINGS

**Datasets.** Following Zügner et al. (2018); Zügner & Günnemann (2019), we validate the proposed approach on four benchmark datasets, including three citation graphs, i.e., Cora, Citeseer, Pubmed,

---

[1] $I_n \in \mathbb{R}^{n \times n}$ denotes the identity matrix. The operator diag acts both on matrices and vectors, and is defined as diag $: \mathbb{R}^{n \times n} \to \mathbb{R}^n$, $\text{diag}(\mathbf{A}) = \sum_{i=1}^n (\mathbf{e}_i^\top \mathbf{A}\mathbf{e}_i)\mathbf{e}_i$, diag $: \mathbb{R}^n \to \mathbb{R}^{n \times n}$, $\text{diag}(\mathbf{a}) = \sum_{i=1}^n (\mathbf{a}^\top\mathbf{e}_i)\mathbf{e}_i\mathbf{e}_i^\top$, with $\mathbf{e}_i \in \mathbb{R}^n$ a one-hot vector with 1 in the $i-$th entry. For matrix input, the main diagonal is extracted. For vector input, its values are placed on the diagonal of a matrix.

and one blog graph, Polblogs . The statistics of the datasets are shown in Appendix H. Note that in the Polblogs graph, node features are not available. In this case, we follow Pro-GNN (Jin et al., 2020) and set the input node features to a $n \times n$ identity matrix.

**Baselines.** We demonstrate the efficacy of CSGNN by comparing it with popular GNNs and defense models, as follows: **GCN** (Kipf & Welling, 2017): Is one of the most commonly used GNN architectures, consisting of feature propagation according to the symmetric normalized Laplacian and channel mixing steps. **GAT** (Veličković et al., 2018): Graph Attention Networks (GAT) employs an attention mechanism to learn edge weights for the feature propagation step. **RGCN** (Zhu et al., 2019): RGCN models node features as samples from Gaussian distributions, and modifies GCN to propagate both the mean and the variance. In the neighborhood aggregation operation, high-variance features are down-weighted to improve robustness. **GCN-Jaccard** (Wu et al., 2019): This is an unsupervised pre-processing method that relies on binary input node features, based on the assumption that the true graph is homophilic. Edges between nodes with features whose Jaccard similarity is below a certain threshold are removed. **GCN-SVD** (Entezari et al., 2020): GCN-SVD is also an unsupervised pre-processing step. Based on the observation that nettack tends to generate high-rank perturbations to the adjacency matrix, it is suggested to truncate the SVD of the adjacency matrix before it is used to train a GNN. **Pro-GNN** (Jin et al., 2020): Pro-GNN attempts to jointly optimize GCN weights and a corrected adjacency matrix using a loss function consisting of a downstream supervised task-related loss function and low-rank and sparsity regularization. In Pro-GNN-fs, an additional feature smoothing regularization is used. **Mid-GCN** (Huang et al., 2023): Mid-GCN modifies the standard GCN architecture to utilize a mid-pass filter, unlike the typical low-pass filter in GCN. **GNNGuard** (Zhang & Zitnik, 2020): GNNGuard modifies message-passing GNNs to include layer-dependent neighbor importance weights in the aggregation step. The neighbor importance weights are designed to favor edges between nodes with similar features, encoding an assumption of homophily. **GRAND** (Feng et al., 2020): In this method, multiple random graph data augmentations are generated, which are then propagated through the GNN. The GNN is trained using a task-related loss and a consistency regularization that encourages similar outputs for the different augmented graphs. **Soft-Median-GDC** (Geisler et al., 2021): This approach first preprocesses the adjacency matrix using graph diffusion convolution (Klicpera et al., 2019), after which a GNN that uses soft median neighborhood aggregation function is trained.

**Training and Evaluation.** We follow the same experimental settings as in Jin et al. (2020). Put precisely, and unless otherwise specified, for each dataset, we randomly choose 10% of the nodes for training, 10% of the nodes for validation, and the remaining 80% nodes for testing. For each experiment, we report the average node classification accuracy of 10 runs. The hyperparameters of all the models are tuned based on the validation set accuracy. In all experiments, the objective function to be minimized is the cross-entropy loss, using the Adam optimizer (Kingma & Ba, 2014). Note, that another benefit of our CSGNN is the use of downstream loss only, compared to other methods that utilize multiple losses to learn adjacency matrix updates. In Appendix I we discuss the hyperparameters of CSGNN. A complexity and runtime discussion is given in Appendix K.

## 5.2 Adversarial Defense Performance

We evaluate the node classification performance of CSGNN against four types of poisoning attacks: (i) non-targeted attack, (ii) targeted attack, (iii) random attack, and, (iv) adaptive attacks. Below we elaborate on the results obtained on each type of attack.

**Robustness to Non-Targeted Adversarial Attacks.** We evaluate the node classification accuracy of our CSGNN and compare it with the baseline methods after using the non-targeted adversarial attack metattack (Zügner & Günnemann, 2019). We follow the publicly available attacks and splits in Jin et al. (2020). We experiment with varying perturbation rates, i.e., the ratio of changed edges, from 0 to 25% with a step size of 5%. We report the average accuracy, as well as the obtained standard deviation over 10 runs in Table 1. The best performing method is highlighted in bold. From Table 1, we can see that except for a few cases, our CSGNN consistently improves or offers on-par performance with other methods.

**Robustness to Targeted Adversarial Attacks.** In this experiment, we use nettack (Zügner et al., 2018) as a targeted attack. Following Zhu et al. (2019), we vary the number of perturbations made on every targeted node from 1 to 5 with a step size of 1. The nodes in the test set with degree larger than

Table 1: Node classification performance (accuracy±std) under a non-targeted attack (metattack) with varying perturbation rates.

| Dataset | Ptb Rate (%) | 0 | 5 | 10 | 15 | 20 | 25 |
|---|---|---|---|---|---|---|---|
| Cora | GCN | 83.50±0.44 | 76.55±0.79 | 70.39±1.28 | 65.10±0.71 | 59.56±2.72 | 47.53±1.96 |
| | GAT | 83.97±0.65 | 80.44±0.74 | 75.61±0.59 | 69.78±1.28 | 59.94±0.92 | 54.78±0.74 |
| | RGCN | 83.09±0.44 | 77.42±0.39 | 72.22±0.38 | 66.82±0.39 | 59.27±0.37 | 50.51±0.78 |
| | GCN-Jaccard | 82.05±0.51 | 79.13±0.59 | 75.16±0.76 | 71.03±0.64 | 65.71±0.89 | 60.82±1.08 |
| | GCN-SVD | 80.63±0.45 | 78.39±0.54 | 71.47±0.83 | 66.69±1.18 | 58.94±1.13 | 52.06±1.19 |
| | Pro-GNN-fs | 83.42±0.52 | 82.78±0.39 | 77.91±0.86 | 76.01±1.12 | 68.78±5.84 | 56.54±2.58 |
| | Pro-GNN | 82.98±0.23 | 82.27±0.45 | 79.03±0.59 | 76.40±1.27 | 73.32±1.56 | 69.72±1.69 |
| | Mid-GCN | **84.61±0.46** | **82.94±0.59** | 80.14±0.86 | 77.77±0.75 | 76.58±0.29 | 72.89±0.81 |
| | CSGNN | 84.12±0.31 | 82.20±0.65 | **80.43±0.74** | **79.32±1.04** | **77.47±1.22** | **74.46±0.99** |
| Citeseer | GCN | 71.96±0.55 | 70.88±0.62 | 67.55±0.89 | 64.52±1.11 | 62.03±3.49 | 56.94±2.09 |
| | GAT | 73.26±0.83 | 72.89±0.83 | 70.63±0.48 | 69.02±1.09 | 61.04±1.52 | 61.85±1.12 |
| | RGCN | 71.20±0.83 | 70.50±0.43 | 67.71±0.30 | 65.69±0.37 | 62.49±1.22 | 55.35±0.66 |
| | GCN-Jaccard | 72.10±0.63 | 70.51±0.97 | 69.54±0.56 | 65.95±0.94 | 59.30±1.40 | 59.89±1.47 |
| | GCN-SVD | 70.65±0.32 | 68.84±0.72 | 68.87±0.62 | 63.26±0.96 | 58.55±1.09 | 57.18±1.87 |
| | Pro-GNN-fs | 73.26±0.38 | 73.09±0.34 | 72.43±0.52 | 70.82±0.87 | 66.19±2.38 | 66.40±2.57 |
| | Pro-GNN | 73.28±0.69 | 72.93±0.57 | 72.51±0.75 | 72.03±1.11 | 70.02±2.28 | 68.95±2.78 |
| | Mid-GCN | 74.17±0.28 | 74.31±0.42 | 73.59±0.29 | 73.69±0.29 | 71.51±0.83 | 69.12±0.72 |
| | CSGNN | **74.93±0.52** | **74.91±0.33** | **73.95±0.35** | **73.82±0.61** | **73.01±0.77** | **72.94±0.56** |
| Polblogs | GCN | 95.69±0.38 | 73.07±0.80 | 70.72±1.13 | 64.96±1.91 | 51.27±1.23 | 49.23±1.36 |
| | GAT | 95.35±0.20 | 83.69±1.45 | 76.32±0.85 | 68.80±1.14 | 51.50±1.63 | 51.19±1.49 |
| | RGCN | 95.22±0.14 | 74.34±0.19 | 71.04±0.34 | 67.28±0.38 | 59.89±0.34 | 56.02±0.56 |
| | GCN-SVD | 95.31±0.18 | 89.09±0.22 | 81.24±0.49 | 68.10±3.73 | 57.33±3.15 | 48.66±9.93 |
| | Pro-GNN-fs | 93.20±0.64 | 93.29±0.18 | 89.42±1.09 | 86.04±2.21 | 79.56±5.68 | 63.18±4.40 |
| | CSGNN | **95.87±0.26** | **95.79±0.15** | **93.21±0.16** | **92.08±0.39** | **90.10±0.37** | **87.37±0.66** |

10 are set as target nodes. Here, we also use the publicly available splits in Jin et al. (2020). The node classification accuracy on target nodes is shown in Figure 2. From the figure, we can observe that when the number of perturbations increases, the performance of CSGNN is better than other methods on the attacked target nodes in most cases.

**Robustness to Random Attacks.** In this experimental setting, we evaluate the performance of CSGNN when the adjacency matrix is attacked by adding random fake edges, from 0% to 100% of the number of edges in the true adjacency matrix, with a step size of 20%. The results are reported in Figure 3. It can be seen, that CSGNN is on par with or better than the considered baselines.

**Robustness to Adaptive Attacks.** We utilize the recently suggested *unit tests* from Mujkanovic et al. (2022). This is a set of perturbed citation datasets, which are notable for the fact that the perturbations

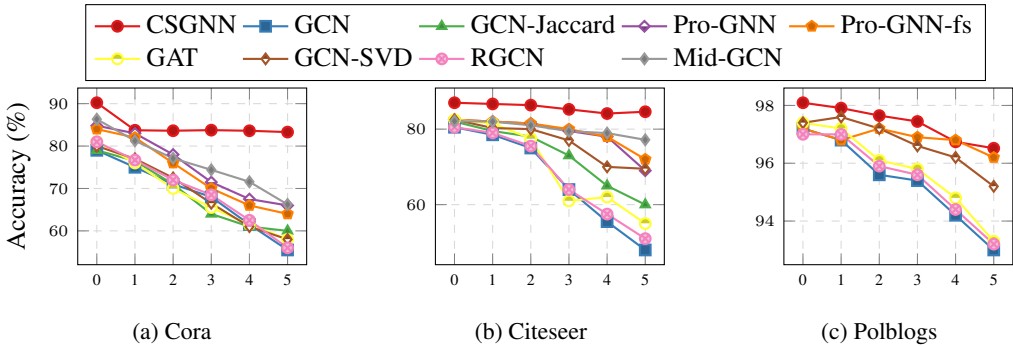

Figure 2: Node classification accuracy (%) under targeted attack with nettack. The horizontal axis describes the number of perturbations per node.

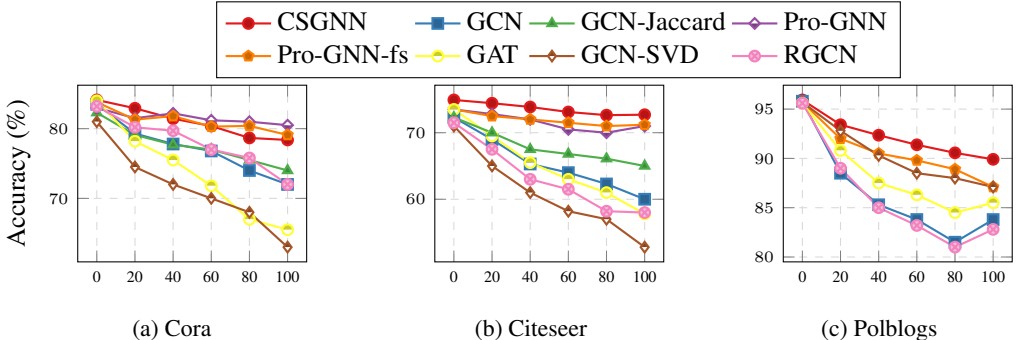

Figure 3: Node classification accuracy (%) under a random adjacency matrix attack. The horizontal axis describes the attack percentage.

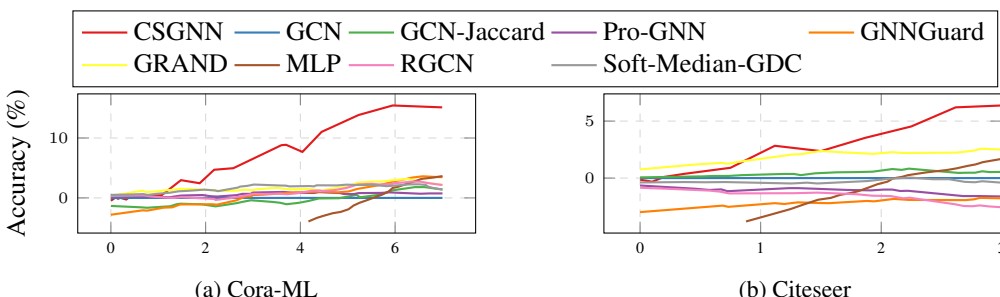

Figure 4: Node classification accuracy (%) using unit-tests from Mujkanovic et al. (2022). Results are relative to a baseline GCN. The horizontal axis shows the attack budget (%).

were not generated using standard attack generation procedures that focus only on attacks like nettack or metattack. Instead, 8 adversarial defense methods were studied. Then, bespoke, adaptive attack methods were designed for each of them. These attack methods were applied to the citation datasets to generate the "unit tests". We experiment with those attacks as they offer a challenging benchmark, that further highlights the contribution of our CSGNN. We present the results in Figure 4, showing the relative performance of CSGNN and other baselines compared to GCN. We see that our CSGNN performs better compared to other considered models. This result further highlights the robustness of CSGNN under different adversarial attack scenarios, on several datasets. In Figure 7 of Appendix J, we also provide absolute performance results, for an additional perspective.

## 6    SUMMARY AND DISCUSSION

In this paper we presented CSGNN, a novel GNN architecture inspired by contractive dynamical systems for graph adversarial defense. Our CSGNN learns a coupled dynamical system that updates both the node features as well as the adjacency matrix to reduce input perturbations impact, thereby defending against graph adversarial attacks. We provide a theoretical analysis of our CSGNN, to gain insights into its characteristics and expected behavior. Our profound experimental study of CSGNN reveals the importance of employing the proposed coupled dynamical system to reduce attack influence on the model's accuracy. Namely, our results verify both the efficacy compared to existing methods, as well as the necessity of each of the dynamical systems in CSGNN. Being the first attempt, to the best of our knowledge, to model both the node features and adjacency matrix through the lens of dynamical systems, we believe that our findings and developments will find further use in graph adversarial defense and attacks, as well as other applications of GNNs.

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

## A    CONTRACTIVE SYSTEMS

This appendix defines continuous contractive dynamical systems and provides a background on the properties of such systems. We focus on contractivity with respect to a norm $\|\cdot\|$ on $\mathbb{R}^k$ induced by an inner product $\langle\cdot,\cdot\rangle : \mathbb{R}^k \times \mathbb{R}^k \to \mathbb{R}$, i.e. so that $\|\mathbf{x}\|^2 = \langle\mathbf{x},\mathbf{x}\rangle$ for every $\mathbf{x} \in \mathbb{R}^k$. This extends thanks to the less restrictive notion of weak pairing considered in Bullo (2023). Following this more general approach, the reasoning extends naturally to the $\ell^1$ norm, i.e., the one used in the case of the dynamical system we propose for the adjacency matrix. Let us consider the dynamical system

$$\begin{cases} \dot{x}(t) = f(x(t)) \in \mathbb{R}^k, \\ x(t_0) = \mathbf{x}_0 \in \mathbb{R}^k. \end{cases} \tag{16}$$

Consider the convex set $\Omega \subset \mathbb{R}^k$. We say $f : \Omega \to \mathbb{R}^k$ satisfies the one-sided Lipschitz inequality on $\Omega$ with constant $\mathrm{osLip}(f) \in \mathbb{R}$ if for every $\mathbf{x}, \mathbf{y} \in \Omega$

$$\langle f(\mathbf{x}) - f(\mathbf{y}), \mathbf{x} - \mathbf{y} \rangle \leq \mathrm{osLip}(f)\|\mathbf{x} - \mathbf{y}\|^2. \tag{17}$$

Additionally, we remark that if $f$ is $\mathrm{Lip}(f)-$Lipschitz continuous, then $\mathrm{osLip}(f) \leq \mathrm{Lip(f)}$ since one has

$$\langle f(\mathbf{x}) - f(\mathbf{y}), \mathbf{x} - \mathbf{y} \rangle \leq \|f(\mathbf{x}) - f(\mathbf{y})\| \cdot \|\mathbf{x} - \mathbf{y}\| \leq \mathrm{Lip}(f)\|\mathbf{x} - \mathbf{y}\|^2.$$

However, while the Lipschitz constant can only be non-negative, the one-sided Lipschitz constant can also be strictly negative. For example, if $f(\mathbf{x}) = -\mathbf{x}$ we get $\mathrm{osLip}(f) = -1$.

**Definition 1** (Contractive dynamical system)**.** *Let $\Omega \subset \mathbb{R}^k$ be a convex set. We say the dynamical system in Equation (16) is strictly contractive on $\Omega$ if it satisfies the one-sided Lipschitz inequality Equation (17) with $\mathrm{osLip}(f) < 0$ for every $\mathbf{x}, \mathbf{y} \in \Omega$, and contractive if $\mathrm{osLip}(f) \leq 0$.*

The motivation behind definition 1 comes from a relatively simple derivation. Let $\mathbf{x}(t)$ and $\mathbf{y}(t)$ be two analytical solutions to Equation (16), respectively with $\mathbf{x}(t_0) = \mathbf{x}_0$ and $\mathbf{y}(t_0) = \mathbf{y}_0$. Then one has

$$\frac{\mathrm{d}}{\mathrm{d}t}\left(\frac{1}{2}\|\mathbf{x}(t) - \mathbf{y}(t)\|^2\right) = \frac{\mathrm{d}}{\mathrm{d}t}\left(\frac{1}{2}\langle\mathbf{x}(t) - \mathbf{y}(t), \mathbf{x}(t) - \mathbf{y}(t)\rangle\right)$$
$$= \langle f(\mathbf{x}(t)) - f(\mathbf{y}(t)), \mathbf{x}(t) - \mathbf{y}(t)\rangle \le \mathrm{osLip}(f)\|\mathbf{x}(t) - \mathbf{y}(t)\|^2.$$

By Gronwall's inequality (Gronwall, 1919), one hence gets

$$\|\mathbf{x}(t) - \mathbf{y}(t)\| \le \|\mathbf{x}_0 - \mathbf{y}_0\| e^{\mathrm{osLip}(f)(t-t_0)}, \quad \forall t \ge t_0.$$

As a consequence, $\|\mathbf{x}(t) - \mathbf{y}(t)\|$ tends to 0 exponentially fast as $t \to +\infty$ if $\mathrm{osLip}(f) < 0$, while it changes in a stable way when $\mathrm{osLip}(f) \le 0$ since

$$\|\mathbf{x}(t) - \mathbf{y}(t)\| \le \|\mathbf{x}_0 - \mathbf{y}_0\|, \quad \forall t \ge t_0.$$

It hence follows that $\mathrm{osLip}(f)$ provides a contraction rate of a generic pair of trajectories, one towards the other. To conclude this section, we focus briefly on the dynamical systems considered for the features. This is of the form

$$\begin{cases} \dot{x}(t) = -\nabla V(x(t)) \\ x(t_0) = \mathbf{x}_0 \end{cases}$$

for a convex function $V : \mathbb{R}^k \to \mathbb{R}$. The properties of convex functions guarantee

$$-\langle\nabla V(\mathbf{x}) - \nabla V(\mathbf{y}), \mathbf{x} - \mathbf{y}\rangle \le 0$$

and hence that $\mathrm{osLip}(-\nabla V) \le 0$. This control on the expansivity of the dynamics is the main motivation for our proposed forward update in Section 4.

## B  CONTRACTIVITY OF THE FEATURE UPDATING RULE

Before proving the contractivity of the feature update rule, we repeat the defintion of the graph gradient operator $\mathcal{G}$, for completeness.

**Definition 2** (Graph gradient operator). *We define the graph gradient operator $\mathcal{G}(\mathbf{A}) : \mathbb{R}^{n \times c} \to \mathbb{R}^{n \times n \times c}$, as follows:*

$$(\mathcal{G}(\mathbf{A})\mathbf{F})_{ijk} = \mathbf{A}_{ij}(\mathbf{F}_{ik} - \mathbf{F}_{jk}), \ i, j \in \{1, \ldots, n\}, k \in \{1, \ldots, c\},$$

*and its transpose $\mathcal{G}(\mathbf{A})^\top : \mathbb{R}^{n \times n \times c} \to \mathbb{R}^{n \times c}$ as*

$$(\mathcal{G}(\mathbf{A})^\top\mathbf{O})_{ik} = \sum_{j=1}^n \left(\mathbf{A}_{ij}\mathbf{O}_{ijk} - \mathbf{A}_{ji}\mathbf{O}_{jik}\right), \ i \in \{1, \ldots, n\}, k \in \{1, \ldots, c\}.$$

Note that in practice we compute it only if the entry $\mathbf{A}_{ij} \ne 0$, and that this gradient operator is just a spatial difference operation applied channel-wise as is common in previous methods (Chamberlain et al., 2021; Eliasof et al., 2021).

We now turn to prove that the node feature update rule is contractive. First of all we notice that $X_l(\mathbf{0}_{n \times c}, \mathbf{A}) = \mathbf{0}_{n \times c}$ for every $\mathbf{A} \in \mathbb{R}^{n \times n}$, and hence since $X_l$ is $L$-Lipschitz for some $L > 0$, we also can conclude

$$\|X_l(\mathbf{F}, \mathbf{A}) - X_l(\mathbf{0}_{n \times c}, \mathbf{A})\|_F = \|X_l(\mathbf{F}, \mathbf{A})\|_F \le L\|\mathbf{F}\|_F. \tag{18}$$

We then introduce the energy

$$\mathcal{E}_\mathbf{A}(\mathbf{F}) = \mathbf{1}_n^\top \gamma(\mathcal{G}(\mathbf{A})\mathbf{F}\mathbf{W}_l))\mathbf{1}_c, \ \gamma'(s) = \sigma(s),$$

where $\mathbf{1}_n \in \mathbb{R}^n$, $\mathbf{1}_c \in \mathbb{R}^n$, are vectors of all ones. This energy is convex in $\mathbf{F}$ since it is obtained by composing convex functions, given that $\sigma$ is non-decreasing. We now write the vectorization of $X_l$:

$$\mathrm{vec}(X_l(\mathbf{F}, \mathbf{A})) = -(\tilde{\mathbf{K}}_l \otimes I_n)\left(\mathbf{W}_l \otimes \mathcal{G}(\mathbf{A})^\top\right)\sigma(\mathrm{vec}(\mathcal{G}(\mathbf{A})\mathbf{F}\mathbf{W}_l))$$
$$= -(\tilde{\mathbf{K}}_l \otimes I_n)\left(\mathbf{W}_l \otimes \mathcal{G}(\mathbf{A})^\top\right)\sigma\left(\left(\mathbf{W}_l^\top \otimes \mathcal{G}(\mathbf{A})\right)\mathrm{vec}(\mathbf{F})\right).$$

Notice that since the gradient of $\mathcal{E}_{\mathbf{A}}$ with respect to $\mathbf{f} := \text{vec}(\mathbf{F})$ writes

$$\nabla_{\mathbf{f}} \mathcal{E}_{\mathbf{A}}(\mathbf{F}) = \left(\mathbf{W}_l \otimes \mathcal{G}(\mathbf{A})^\top\right) \sigma\left(\left(\mathbf{W}_l^\top \otimes \mathcal{G}(\mathbf{A})\right) \mathbf{f}\right),$$

we can express $\text{vec}(X_l(\mathbf{A}, \mathbf{F}))$ in the simpler form

$$\hat{X}_l(\mathbf{F}, \mathbf{A}) := \text{vec}(X_l(\mathbf{F}, \mathbf{A})) = -(\tilde{\mathbf{K}}_l \otimes I_n)\nabla_{\mathbf{f}} \mathcal{E}_{\mathbf{A}}(\mathbf{F}).$$

This allows us to prove both Theorems 1 and 2 for two interesting configurations. First, we notice that if $\tilde{\mathbf{K}}_l$ is positive definite, when $\mathbf{F} \neq \mathbf{0}_{n \times c}$ and $\nabla_{\mathbf{f}} \mathcal{E}_{\mathbf{A}}(\mathbf{F}) \neq \mathbf{0}_{n \times c}$, we have

$$\nabla_{\mathbf{f}} \mathcal{E}_{\mathbf{A}}(\mathbf{F})^\top \text{vec}(X_l(\mathbf{A}, \mathbf{F})) \leq \lambda_{\max}(-\tilde{\mathbf{K}}_l)\|\nabla_{\mathbf{f}} \mathcal{E}_{\mathbf{A}}(\mathbf{F})\|_2^2 \leq L^2 \lambda_{\max}(-\tilde{\mathbf{K}}_l)\|\mathbf{F}\|_F^2 < 0.$$

Here , with $\lambda_{\max}(-\tilde{\mathbf{K}}_l)$ we denote the maximum eigenvalue of $-\tilde{\mathbf{K}}_l$, which is negative by definition of positive definite matrices. We then have

$$\mathcal{E}_{\mathbf{A}}(\Psi_{X_l}^{h_l}(\mathbf{F}, \mathbf{A})) \leq \mathcal{E}_{\mathbf{A}}(\mathbf{F})$$

for small enough $h_l > 0$ since $\hat{X}_l$ locally provides a descent direction for $\mathcal{E}_{\mathbf{A}}$. This result guarantees that the updates $\mathbf{F}^{(l)}$ will remain bounded.

To prove Theorem 2, we first notice that

$$\|\mathbf{P}\|_F = \|\text{vec}(\mathbf{P})\|_2 \quad \forall \mathbf{P} \in \mathbb{R}^{n \times c},$$

and thus prove the result for the vectorization of $X_l(\mathbf{A}, \mathbf{F})$. We focus on the case $\mathbf{K}_l = \lambda I_c$, $\lambda > 0$, which is the one tested in the experiments of Appendix J. Given that in this case $\text{vec}(X_l(\mathbf{F}, \mathbf{A})) = -\lambda \nabla_{\mathbf{f}}(\mathcal{E}_{\mathbf{A}}(\mathbf{F}, \mathbf{A}))$, we can immediately conclude. Indeed, we can apply the results in Sherry et al. (2023), for example, to prove the desired result for every $\mathbf{W}_l \in \mathbb{R}^{n \times n}$. This is just a direct consequence of the properties of convex functions with Lipschitz gradient.

## C  PROOFS FOR THE CONTRACTIVITY OF THE ADJACENCY MATRIX UPDATES

This section aims to provide a detailed proof of Theorem 3. This is divided into various steps. We first provide the vectorized version of such theorem, which we then prove. The theorem then follows directly by definition of the vectorized $\ell^1$ norm.

Let $M(\mathbf{A})$ be defined as in Equation (14). It is clear that $M$ is linear in $\mathbf{A} \in \mathbb{R}^{n \times n}$, and thus that there exists a matrix $\mathbf{T} \in \mathbb{R}^{n^2 \times n^2}$ such that

$$\text{vec}\left(M(\mathbf{A}^{(l)})\right) = \mathbf{T} \, \text{vec}(\mathbf{A}^{(l)}). \tag{19}$$

We now characterize the explicit expression of such $\mathbf{T}$.

**Theorem 4.** *The matrix $\mathbf{T}$ can be written as follows:*

$$
\begin{aligned}
\mathbf{T} = {} & k_1 I_{n^2} + k_2 \sum_{i=1}^n (\mathbf{e}_i \mathbf{e}_i^\top) \otimes (\mathbf{e}_i \mathbf{e}_i^\top) + \frac{k_3}{2n}\left(\mathbf{1}_n \mathbf{1}_n^\top \otimes I_n + I_n \otimes \mathbf{1}_n \mathbf{1}_n^\top\right) \\
& + k_4 \sum_{i=1}^n (\mathbf{e}_i \mathbf{1}_n^\top) \otimes (\mathbf{e}_i \mathbf{e}_i^\top) + \frac{k_5}{n^2}(\mathbf{1}_n \mathbf{1}_n^\top) \otimes (\mathbf{1}_n \mathbf{1}_n^\top) \\
& + \frac{k_6}{n} \sum_{i=1}^n (\mathbf{e}_i \mathbf{1}_n^\top) \otimes (\mathbf{e}_i \mathbf{1}_n^\top) + \frac{k_7}{n^2} \sum_{i=1}^n (\mathbf{1}_n \mathbf{e}_i^\top) \otimes (\mathbf{1}_n \mathbf{e}_i^\top) \\
& + \frac{k_8}{n} \sum_{i,j=1}^n (\mathbf{e}_j \mathbf{e}_i^\top) \otimes (\mathbf{e}_j \mathbf{e}_i^\top) + \frac{k_9}{2n} \sum_{i=1}^n \left((\mathbf{1}_n \mathbf{e}_i^\top) \otimes (\mathbf{e}_i \mathbf{e}_i^\top) + (\mathbf{e}_i \mathbf{e}_i^\top) \otimes (\mathbf{1}_n \mathbf{e}_i^\top)\right)
\end{aligned} \tag{20}
$$

*where $\mathbf{e}_i \in \mathbb{R}^{n^2}$ is the $i-th$ element of the canonical basis.*

*Proof.* We do it by focusing on the separate 9 pieces defining $\mathbf{T}$. The main result needed to complete the derivation is that

$$\text{vec}(\mathbf{ABC}) = (\mathbf{C}^\top \otimes \mathbf{A}) \text{vec}(\mathbf{B}). \tag{21}$$

We omit the part with $k_1$ since it is trivial. Let us start with

$$\text{diag}(\text{diag}(\mathbf{A})) = \sum_{i=1}^{n}(\mathbf{e}_i^\top A \mathbf{e}_i)\mathbf{e}_i\mathbf{e}_i^\top = \sum_{i=1}^{n}\mathbf{e}_i(\mathbf{e}_i^\top A \mathbf{e}_i)\mathbf{e}_i^\top$$
$$= \sum_{i=1}^{n}(\mathbf{e}_i\mathbf{e}_i^\top)A(\mathbf{e}_i\mathbf{e}_i^\top)$$

which allows us to conclude the proof by the linearity of the vec operator, and Equation (21). Moving to the $k_3$ part we have that by Equation (21)

$$\text{vec}\left(\mathbf{A}\mathbf{1}_n\mathbf{1}_n^\top + \mathbf{1}_n\mathbf{1}_n^\top\mathbf{A}\right) = (\mathbf{1}_n\mathbf{1}_n^\top \otimes I_n + I_n \otimes \mathbf{1}_n\mathbf{1}_n^\top)\text{vec}(A).$$

The $k_4$ part can be rewritten as

$$\text{diag}(\mathbf{A}\mathbf{1}_n) = \sum_{i=1}^{n}(\mathbf{e}_i^\top\mathbf{A}\mathbf{1}_n)\mathbf{e}_i\mathbf{e}_i^\top = \sum_{i=1}^{n}\mathbf{e}_i\mathbf{e}_i^\top\mathbf{A}\mathbf{1}_n\mathbf{e}_i^\top$$

which gives

$$\text{vec}(\text{diag}(\mathbf{A}\mathbf{1}_n)) = \left(\sum_{i=1}^{n}(\mathbf{e}_i\mathbf{1}_n^\top) \otimes (\mathbf{e}_i\mathbf{e}_i^\top)\right)\text{vec}(\mathbf{A}).$$

Term for $k_5$ follows immediately from Equation (21), while for $k_6$ we can write

$$(\mathbf{1}_n^\top\mathbf{A}\mathbf{1}_n)I_n = (\mathbf{1}_n^\top\mathbf{A}\mathbf{1}_n)\sum_{i=1}^{n}\mathbf{e}_i\mathbf{e}_i^\top = \sum_{i=1}^{n}\mathbf{e}_i(\mathbf{1}_n^\top\mathbf{A}\mathbf{1}_n)\mathbf{e}_i^\top$$

which implies the desired expression. For $k_7$ it holds

$$(\mathbf{1}_n^\top\text{diag}(\mathbf{A}))\mathbf{1}_n\mathbf{1}_n^\top = \sum_{i=1}^{n}(\mathbf{e}_i^\top\mathbf{A}\mathbf{e}_i)\mathbf{1}_n\mathbf{1}_n^\top = \sum_{i=1}^{n}\mathbf{1}_n\mathbf{e}_i^\top\mathbf{A}\mathbf{e}_i\mathbf{1}_n^\top$$

which allows us to conclude. We now move to

$$(\mathbf{1}_n^\top\text{diag}(\mathbf{A}))I_n = \sum_{i=1}^{n}(\mathbf{e}_i^\top\mathbf{A}\mathbf{e}_i)\sum_{j=1}^{n}\mathbf{e}_j\mathbf{e}_j^\top = \sum_{i,j=1}^{n}\mathbf{e}_j\mathbf{e}_i^\top\mathbf{A}\mathbf{e}_i\mathbf{e}_j^\top$$

and hence

$$\text{vec}((\mathbf{1}_n^\top\text{diag}(\mathbf{A}))I_n) = \left(\sum_{i,j=1}^{n}(\mathbf{e}_j\mathbf{e}_i^\top) \otimes (\mathbf{e}_j\mathbf{e}_i^\top)\right)\text{vec}(\mathbf{A}).$$

We consider now the term multiplying $k_9$ which writes

$$\text{diag}(\mathbf{A})\mathbf{1}_n^\top + \mathbf{1}_n\text{diag}(\mathbf{A})^\top = \sum_{i=1}^{n}(\mathbf{e}_i^\top\mathbf{A}\mathbf{e}_i)\mathbf{e}_i\mathbf{1}_n^\top + \mathbf{1}_n\sum_{i=1}^{n}(\mathbf{e}_i^\top\mathbf{A}\mathbf{e}_i)\mathbf{e}_i^\top,$$

$$\text{vec}(\text{diag}(\mathbf{A})\mathbf{1}_n^\top + \mathbf{1}_n\text{diag}(\mathbf{A})^\top) = \left(\sum_{i=1}^{n}(\mathbf{1}_n\mathbf{e}_i^\top) \otimes (\mathbf{e}_i\mathbf{e}_i^\top) + (\mathbf{e}_i\mathbf{e}_i^\top) \otimes (\mathbf{1}_n\mathbf{e}_i^\top)\right)\text{vec}(\mathbf{A}),$$

and concludes the proof. $\qquad\square$

We now want to evaluate how large can $h_l > 0$ be so that the condition $\|D\Psi_{\hat{Y}_l}^{h_l}\|_1 \le 1$ is satisfied. Let us note here that the norm under consideration is the *matrix* $\ell^1$ norm, not the usual *vector* $\ell^1$ norm which has previously occurred in the main text. Recall that the matrix $\ell^1$ norm of a matrix $\mathbf{T} \in \mathbb{R}^{n^2 \times n^2}$ is given as the maximum of the absolute column sums:

$$\|\mathbf{T}\|_1 = \max_{1 \le j \le n^2}\sum_{i=1}^{n^2}|\mathbf{T}_{ij}|.$$

**Theorem 5.** *The matrix* $\mathbf{T} - k_1 I_{n^2}$, *with* $\mathbf{T}$ *defined in Equation* (20), *has* $\ell^1$ *norm bounded by* $\sum_{i=2}^{9} |k_i|$.

*Proof.* Given that one can bound the norm of the sum with the sum of the norms, it is enough to show that the norms of all the contributions in Equation (20) can be bounded by the absolute value of the respective constant $k_i$. The proof follows from multiplying from the left every term by $\mathbf{1}_{n^2}$, and using the linearity of the sum. □

For compactness, we now denote $\text{vec}(Y_l(\mathbf{A}))$ with $\hat{Y}_l(\mathbf{a})$, where $\mathbf{a} = \text{vec}(\mathbf{A})$. Furthermore, the map $\Psi^{h_l}_{\hat{Y}_l}$ is defined as $\Psi^{h_l}_{\hat{Y}_l}(\mathbf{a}) := \text{vec}(\Psi^{h_l}_{Y_l}(\mathbf{A}))$.

**Theorem 6.** *Let* $\alpha \le 0$, $\sigma : \mathbb{R} \to \mathbb{R}$ *be a Lipschitz continuous function, with* $\sigma'(s) \in [0, 1]$. *Then if*

$$0 \le h_l \le \frac{2}{2\sum_{i=2}^{9} |k_i| - \alpha},$$

*the explicit Euler step*

$$\mathbf{a}^{(l)} = \Psi^{h_l}_{\hat{Y}_l}(\mathbf{a}^{(l-1)}) := \mathbf{a}^{(l-1)} + h_l \sigma\left(\mathbf{T}\mathbf{a}^{(l-1)}\right), \quad k_1 = \left(\alpha - \sum_{i=2}^{9} |k_i|\right), \qquad (22)$$

*is contractive in the* $\ell^1$ *norm.*

For the proof and the successive derivations, we denote with $D\hat{Y}_l(\mathbf{a}) \in \mathbb{R}^{n^2 \times n^2}$ the Jacobian matrix of the vector field $\hat{Y}_l : \mathbb{R}^{n^2} \to \mathbb{R}^{n^2}$, having entries

$$(D\hat{Y}_l(\mathbf{a}))_{ij} = \frac{\partial(\hat{Y}_l(\mathbf{a}))_i}{\partial \mathbf{a}_j}. \qquad (23)$$

The Jacobian matrix is needed because, for functions that are almost everywhere continuously differentiable, the contractivity condition is equivalent to

$$\|D\Psi^{h_l}_{\hat{Y}_l}(\mathbf{a})\|_1 \le 1 \qquad (24)$$

almost everywhere.

*Proof.* For compactness, we drop the superscript and denote $\mathbf{a}^{(l-1)}$ as $\mathbf{a}$, since the provided estimates are independent of the evaluation point. The map $\Psi^{h_l}_{\hat{Y}_l}$ is differentiable almost everywhere and hence the result simplifies to proving

$$\|D\Psi^{h_l}_{\hat{Y}_l}(\mathbf{a})\|_1 \le 1.$$

We thus compute

$$D\Psi^{h_l}_{\hat{Y}_l}(\mathbf{a}) = I_{n^2} + h_l \text{diag}\left(\sigma'(\mathbf{T}\mathbf{a})\right)\mathbf{T}.$$

To simplify the proof, we introduce the matrix $\mathbf{S} = \mathbf{T} - k_1 I_{n^2}$. This means that the update in Equation (22) can be written as

$$\Psi^{h_l}_{\hat{Y}_l}(\mathbf{a}) = \mathbf{a} + h_l \sigma\left(\mathbf{S}\mathbf{a} + \left(\alpha - \sum_{i=2}^{9} |k_i|\right)\mathbf{a}\right).$$

We call $d_1, \ldots, d_{n^2}$ the diagonal entries of the matrix $\text{diag}(\sigma'(\cdot))$. It follows

$$(D\Psi^{h_l}_{\hat{Y}_l}(\mathbf{a}))_{ii} = 1 + h_l d_i \left(\mathbf{S}_{ii} + \left(\alpha - \sum_{i=2}^{9} |k_i|\right)\right)$$

$$(D\Psi^{h_l}_{\hat{Y}_l}(\mathbf{a}))_{ji} = h_l d_j \mathbf{S}_{ji}.$$

If $(D\Psi^{h_l}_{\hat{Y}_l}(\mathbf{a}))_{ii} \ge 0$, one gets

$$\sum_{j=1}^{n^2} |(D\Psi^{h_l}_{\hat{Y}_l}(\mathbf{a}))_{ji}| = 1 + h_l d_i \left(\mathbf{S}_{ii} + \left(\alpha - \sum_{i=2}^{9} |k_i|\right)\right) + h_l \sum_{j \ne i} d_j |\mathbf{S}_{ji}| \le 1.$$

This inequality holds whenever

$$d_i \mathbf{S}_{ii} + \sum_{j \neq i} d_j |\mathbf{S}_{ji}| + \left( \alpha - \sum_{i=2}^{9} |k_i| \right) \leq 0,$$

which is always true since

$$d_i \mathbf{S}_{ii} + \sum_{j \neq i} d_j |\mathbf{S}_{ji}| + \left( \alpha - \sum_{i=2}^{9} |k_i| \right) \leq d_i |\mathbf{S}_{ii}| + \sum_{j \neq i} d_j |\mathbf{S}_{ji}| + \left( \alpha - \sum_{i=2}^{9} |k_i| \right)$$

$$\leq \|\mathbf{S}\|_1 - \sum_{i=2}^{9} |k_i| + \alpha \leq \alpha \leq 0.$$

It hence only remains to study the case $(D\Psi^{h_l}_{\hat{Y}_l}(\mathbf{a}))_{ii} < 0$, which leads to

$$\sum_{j=1}^{n^2} |(D\Psi^{h_l}_{\hat{Y}_l}(\mathbf{a}))_{ji}| = -1 - h_l d_i \left( \mathbf{S}_{ii} + \left( \alpha - \sum_{i=2}^{9} |k_i| \right) \right) + h_l \sum_{j \neq i} d_j |\mathbf{S}_{ji}| \leq 1.$$

We move to a more stringent condition which is given by bounding $-h_l d_i \mathbf{S}_{ii} \leq h_l d_i |\mathbf{S}_{ii}| \leq h_l |\mathbf{S}_{ii}|$, so that we get

$$\sum_{j=1}^{n^2} |(D\Psi^{h_l}_{\hat{Y}_l}(\mathbf{a}))_{ji}| \leq -1 + h_l |\mathbf{S}_{ii}| - h_l d_i \left( \alpha - \sum_{i=2}^{9} |k_i| \right) + h_l \sum_{j \neq i} |\mathbf{S}_{ji}|$$

$$\leq -1 + h_l \|\mathbf{S}\|_1 - h_l d_i \left( \alpha - \sum_{i=2}^{9} |k_i| \right) \leq 1.$$

This holds true when

$$h_l \leq \frac{2}{\|\mathbf{S}\|_1 - d_i \alpha + d_i \sum_{i=2}^{9} |k_i|}.$$

Now since $\|\mathbf{S}\|_1 \leq \sum_{i=2}^{9} |k_i|$, and $-d_i \alpha \in [0, -\alpha]$, we have

$$\frac{2}{2 \sum_{i=1}^{9} |k_i| - \alpha} \leq \frac{2}{\|\mathbf{S}\|_1 - d_i \alpha + d_i \sum_{i=1}^{9} |k_i|},$$

which allows to conclude that if

$$0 \leq h_l \leq \frac{2}{2 \sum_{i=2}^{9} |k_i| - \alpha}$$

then the contractivity condition is satisfied. □

By definition of the vectorized $\ell^1$ norm, one can hence conclude that Theorem 6 is equivalent to Theorem 3.

## D  PROOF OF THE CONTRACTIVITY OF THE COUPLED DYNAMICAL SYSTEM

In this section, we work with the coupled system

$$\begin{cases} \dot{F}(t) = -\mathcal{G}(A(t))^\top \sigma(\mathcal{G}(A(t))F(t)W(t))W(t)^\top \\ \dot{A}(t) = \sigma(M(A(t))), \end{cases} \tag{25}$$

where $M$ is defined as in Section 4.3, and hence defines an equivariant system which is also contractive in vectorized $\ell^1$ norm. Consider only the bounded time interval $t \in [0, \bar{T}]$, where we assume that the graph defined by $t \mapsto A(t)$ starts and remains connected. This guarantees $\mathcal{G}(A(t))F(t) = 0$ if and only if $F(t) \equiv F(0)$, and $F(0)$ has coinciding rows. We suppose this does not happen at time $t = 0$.

We assume that the activation function is $\sigma = \text{LeakyReLU}$. In the adjacency dynamical system, we set $\alpha < 0$. Furthermore, let $W_l$ be non-singular. We now show that Equation (25) provides a contractive continuous dynamical system in a suitable norm.

The focus is now on the time-independent case, i.e. only on $X_l$ and $Y_l$, since the more general case follows naturally. We thus specify the expression in Equation (25) for this setting:

$$\begin{cases} \dot{F}(t) = -\mathcal{G}(A(t))^\top \sigma(\mathcal{G}(A(t))F(t)W_l)W_l^\top =: X_l(F(t), A(t)) \\ \dot{A}(t) = \sigma(M(A(t))) =: Y_l(A(t)). \end{cases} \tag{26}$$

Using the concepts described in Appendix A, it is possible to prove that the two separate equations are strictly contracting in their respective norms, that is

$$\|F(t) - F_*(t)\|_F \le e^{-\nu_1 t}\|\mathbf{F}^{(0)} - \mathbf{F}_*^{(0)}\|_F, \ \nu_1 > 0, \ t \in [0, \bar{T}]$$

$$\|\text{vec}(A(t)) - \text{vec}(A_*(t))\|_1 \le e^{-\nu_2 t}\|\text{vec}(\mathbf{A}^{(0)}) - \text{vec}(\mathbf{A}_*^{(0)})\|_1, \ \nu_2 > 0, \ t \in [0, \bar{T}],$$

where $(F(t), A(t))$ and $(F_*(t), A_*(t))$ are solutions of Equation (26), when considered separately, and starting respectively at $(\mathbf{F}^{(0)}, \mathbf{A}^{(0)})$ and $(\mathbf{F}_*^{(0)}, \mathbf{A}_*^{(0)})$. For more details on this result see Bullo (2023). We remark that in the first inequality, the $\mathbf{A}^{(0)}$ matrix is seen as a parameter, and hence does not evolve. These two conditions thus say that the two systems are strictly contracting when considered separately. In Sontag (2010), the author shows that when these conditions hold and the mixed Jacobian matrix

$$J(\mathbf{F}, \mathbf{A}) = \frac{\partial \, \text{vec}(X_l(\mathbf{F}, \mathbf{A}))}{\partial \, \text{vec}(\mathbf{A})} \in \mathbb{R}^{nc \times n^2} \tag{27}$$

is bounded in a suitable norm, there is a pair of constants $m_1, m_2 > 0$ such that the system in Equation (25) is contractive with respect to the weighted norm

$$d_{m_1, m_2}((F(t), A(t)), (F_*(t), A_*(t))) := m_1\|\mathbf{F}^{(0)} - \mathbf{F}_*^{(0)}\|_F + m_2\|\text{vec}(\mathbf{A}^{(0)}) - \text{vec}(\mathbf{A}_*^{(0)})\|_1.$$

Now, $(F(t), A(t))$ and $(F_*(t), A_*(t))$ are solutions of Equation (26) considered as a coupled system, i.e. solving jointly the two equations. More precisely, the mixed Jacobian matrix has to be bounded in the operator norm $\|\cdot\|_{1,2}$ defined by $\|\cdot\|_2$ and $\|\cdot\|_1$, i.e.

$$\|J(\mathbf{F}, \mathbf{A})\|_{1,2} := \max_{\substack{\mathbf{v} \in \mathbb{R}^{n^2}, \\ \|\mathbf{v}\|_1 = 1}} \|J(\mathbf{F}, \mathbf{A})\mathbf{v}\|_2 < +\infty$$

for every $(\mathbf{F}, \mathbf{A}) \in \Omega \subset \mathbb{R}^{n \times c} \times \mathbb{R}^{n \times n}$, with $\Omega$ closed and convex. Notice also that since $Y_l(\mathbf{0}_{n \times n}) = \mathbf{0}_{n \times n}$, and such system is contractive, it follows that any solution of Equation (25) has $A(t)$ which is bounded uniformly in time by the norm of the initial condition $\|\text{vec}(\mathbf{A}^{(0)})\|_1$. Similarly, we notice that for every $\mathbf{A}$, one also has $X_l(\mathbf{0}_{n \times c}, \mathbf{A}) = \mathbf{0}_{n \times c}$ and hence also $\|F(t)\|_F$ is bounded by $\|\mathbf{F}^{(0)}\|_F$. To compute the Jacobian and its norm we first define $\mathcal{G}$ and $\mathcal{G}^T$, as matrix operators, by specifying their components when acting on generic $\mathbf{F} \in \mathbb{R}^{n \times c}$ and $\mathbf{O} \in \mathbb{R}^{n \times n \times c}$:

$$(\mathcal{G}(\mathbf{A})\mathbf{F})_{ijk} = \mathbf{A}_{ij}(\mathbf{F}_{ik} - \mathbf{F}_{jk}), \ i, j \in \{1, \ldots, n\}, \ k \in \{1, \ldots c\},$$

$$(\mathcal{G}(\mathbf{A})^\top \mathbf{O})_{ik} = \sum_{j=1}^{n} (\mathbf{A}_{ij}\mathbf{O}_{ijk} - \mathbf{A}_{ji}\mathbf{O}_{jik}), \ i \in \{1, \ldots, n\}, \ k \in \{1, \ldots c\}.$$

Here, $\mathcal{G}(\mathbf{A})^\top \mathbf{O}$ is obtained thanks to the relation $\text{vec}(\mathcal{G}(\mathbf{A})\mathbf{F})^\top \text{vec}(\mathbf{O}) = \text{vec}(\mathbf{F})^\top \text{vec}(\mathcal{G}(\mathbf{A})^\top \mathbf{O})$.

We remark that $\sigma$ is of class $C^1$ almost everywhere, while all the other functions composed in order to define $X_l$ are smooth in their arguments. This immediately allows us to conclude that the Jacobian $J$ is bounded in norm on compact sets. Recalling that $(F(t), A(t))$ evolve in the compact set $\{(\mathbf{F}, \mathbf{A}) \in \mathbb{R}^{n \times c} \times \mathbb{R}^{n \times n} : \|\mathbf{F}\|_F \le \|\mathbf{F}^{(0)}\|_F, \ \|\text{vec}(\mathbf{A})\|_1 \le \|\text{vec}(\mathbf{A}^{(0)})\|_1\}$, we can conclude that $J$ is bounded. Thus, one can conclude the contractivity of the coupled dynamics in Equation (25) for a suitable norm defined as

$$d_{m_1, m_2}((\mathbf{F}, \mathbf{A}), (\mathbf{F}_*, \mathbf{A}_*)) := m_1\|\mathbf{F} - \mathbf{F}_*\|_F + m_2\|\text{vec}(\mathbf{A}) - \text{vec}(\mathbf{A}_*)\|_1$$

for $m_1, m_2 > 0$ large enough.

# E    RELATED WORKS: ADVERSARIAL ROBUSTNESS VIA DYNAMICAL SYSTEMS AND LIPSCHITZ REGULARITY

The approach of improving the robustness of neural networks through Lipschitz constraints and techniques typical of the stability theory of dynamical systems has attracted much interest in recent years. This research direction has been investigated especially for classification tasks based on structured grids, i.e. images. For completeness in the presentation, we mention a few relevant contributions based on these insights and briefly comment on them. In Yang et al. (2023), the authors work with evasion attacks and start by observing that not all input perturbations lead to changes in the predicted class. They then provide robustness guarantees based on the stability theory of dynamical systems. In Kang et al. (2021), the authors again propose to enhance the robustness of a neural network based on Lyapunov stability. More precisely, they design a loss function promoting the closeness of output predictions to stable equilibria of the differential equation ruling the network, and also that these equilibria are as far as possible when different classes are considered. In Huang et al. (2022), the authors work with neural-controlled ODEs and introduce a framework based on forward invariance. They propose a strategy to turn a desired function into a Lyapunov function for the ODE driving the neural network, hence having its sublevel sets be forward invariant. This technique and some sampling strategies allow them to get certifiably robust models. In Sherry et al. (2023); Zakwan et al. (2022), the authors exploit the connection between ResNet architectures and numerical methods, as in this manuscript, to design contractive residual neural networks based on the discretisation of contractive dynamical systems. The experimental setup of both these works is again based on evasion attacks for image-based classification problems. Together with these mentioned works, which rely on dynamical systems theory, a big body of literature proposes to constrain the Lipschitz constant of the network as a way of reducing its sensitivity to input perturbations, see Tsuzuku et al. (2018a); Pauli et al. (2021); Liu et al. (2021); Huster et al. (2019).

# F    LIPSCHITZ CONSTANT OF THE MAP $\mathcal{D}$

We now consider the bound on the Lipschitz constant of the network part $\mathcal{D}$ obtained by composing explicit Euler steps of dynamical systems. We consider the setting described in Appendix D, and assume all the steps $h_1, \ldots, h_L$ to be small enough so that these explicit Euler steps are also contractive, see Theorems 2 and 3. These maps are contractive when considered in isolation, but, in general, their composition is not. We recall the system of differential equations to consider:

$$\begin{cases} \dot{F}(t) = X(t, F(t), A(t)) \\ \dot{A}(t) = Y(t, A(t)). \end{cases} \tag{28}$$

To specify the Lipschitz constant of the map $\mathcal{D}$, we introduce the two following maps:

$$X_{l,\mathbf{A}} : \mathbb{R}^{n \times c} \to \mathbb{R}^{n \times c}, \ X_{l,\mathbf{A}}(\mathbf{F}) := X_l(\mathbf{F}, \mathbf{A}) = X(\tau_{l-1}, \mathbf{F}, \mathbf{A}), \tag{29}$$

$$X_{l,\mathbf{F}} : \mathbb{R}^{n \times n} \to \mathbb{R}^{n \times c}, \ X_{l,\mathbf{F}}(\mathbf{A}) := X_l(\mathbf{F}, \mathbf{A}) = X(\tau_{l-1}, \mathbf{F}, \mathbf{A}). \tag{30}$$

Let us first recall the expression for $\mathcal{D}$, which is

$$\mathcal{D} = \mathcal{D}_L \circ \ldots \circ \mathcal{D}_1,$$

where

$$\mathcal{D}_l((\mathbf{F}^{(l-1)}, \mathbf{A}^{(l-1)})) = \left( \Psi^{h_l}_{X_{l,\mathbf{A}^{(l-1)}}}(\mathbf{F}^{(l-1)}), \Psi^{h_l}_{Y_l}(\mathbf{A}^{(l-1)}) \right), \quad l = 1, \ldots, L.$$

We consider the two pairs of initial conditions $(\mathbf{F}^{(0)}, \mathbf{A}^{(0)})$, $(\mathbf{F}_*^{(0)}, \mathbf{A}_*^{(0)})$, and update first the $\mathbf{F}^{(0)}$ component with $\Psi_{X_{1,\mathbf{A}^{(0)}}}^{h_1}$, and $\mathbf{F}_*^{(0)}$ with $\Psi_{X_{1,\mathbf{A}_*^{(0)}}}^{h_1}$ to get

$$\mathbf{F}^{(1)} = \Psi_{X_{1,\mathbf{A}^{(0)}}}^{h_1}(\mathbf{F}^{(0)}) = \mathbf{F}^{(0)} + h_1 X_{1,\mathbf{A}^{(0)}}(\mathbf{F}^{(0)}) \tag{31}$$

$$\mathbf{F}_*^{(1)} = \Psi_{X_{1,\mathbf{A}_*^{(0)}}}^{h_1}(\mathbf{F}_*^{(0)}) = \mathbf{F}_*^{(0)} + h_1 X_{1,\mathbf{A}_*^{(0)}}(\mathbf{F}_*^{(0)}) \tag{32}$$

$$\|\mathbf{F}^{(1)} - \mathbf{F}_*^{(1)}\|_F = \|\Psi_{X_{1,\mathbf{A}^{(0)}}}^{h_1}(\mathbf{F}^{(0)}) + (\Psi_{X_{1,\mathbf{A}_*^{(0)}}}^{h_1}(\mathbf{F}^{(0)}) - \Psi_{X_{1,\mathbf{A}_*^{(0)}}}^{h_1}(\mathbf{F}^{(0)})) - \Psi_{X_{1,\mathbf{A}_*^{(0)}}}^{h_1}(\mathbf{F}_*^{(0)})\|_F$$

$$\le \|\Psi_{X_{1,\mathbf{A}^{(0)}}}^{h_1}(\mathbf{F}^{(0)}) - \Psi_{X_{1,\mathbf{A}_*^{(0)}}}^{h_1}(\mathbf{F}^{(0)})\|_F + \|\mathbf{F}^{(0)} - \mathbf{F}_*^{(0)}\|_F$$

$$\le h_1 \|X_{1,\mathbf{F}^{(0)}}(\mathbf{A}^{(0)}) - X_{1,\mathbf{F}^{(0)}}(\mathbf{A}_*^{(0)})\|_F + \varepsilon_1$$

$$\le h_1 \mathrm{Lip}(X_{1,\mathbf{F}^{(0)}}) \| \mathrm{vec}(\mathbf{A}^{(0)}) - \mathrm{vec}(\mathbf{A}_*^{(0)})\|_1 + \varepsilon_1.$$

$$\le h_1 \mathrm{Lip}(X_{1,\mathbf{F}^{(0)}}) \varepsilon_2 + \varepsilon_1. \tag{33}$$

By $\mathrm{Lip}(X_{1,\mathbf{F}^{(0)}})$ we refer to the Lipschitz constant of the map $X_{1,\mathbf{F}^{(0)}} : \mathbb{R}^{n \times n} \to \mathbb{R}^{n \times c}$, where the first space has the vectorized $\ell^1$ norm, and the second has the Frobenius norm.

Then one can update the adjacency matrices $\mathbf{A}^{(0)}$ and $\mathbf{A}_*^{(0)}$ to

$$\mathbf{A}^{(1)} = \Psi_{Y_1}^{h_1}(\mathbf{A}^{(0)}), \ \ \mathbf{A}_*^{(1)} = \Psi_{Y_1}^{h_1}(\mathbf{A}_*^{(0)}),$$

for which we have already proven $\| \mathrm{vec}(\mathbf{A}^{(1)}) - \mathrm{vec}(\mathbf{A}_*^{(1)})\|_1 \le \varepsilon_2$. To get the general form of the Lipschitz constant of $\mathcal{D}$, we update again the features so it is easier to generalize:

$$\mathbf{F}^{(2)} = \Psi_{X_{2,\mathbf{A}^{(1)}}}^{h_2}(\mathbf{F}^{(1)}) = \mathbf{F}^{(1)} + h_2 X_{2,\mathbf{A}^{(1)}}(\mathbf{F}^{(1)}) \tag{34}$$

$$\mathbf{F}_*^{(2)} = \Psi_{X_{2,\mathbf{A}_*^{(1)}}}^{h_2}(\mathbf{F}_*^{(1)}) = \mathbf{F}_*^{(1)} + h_2 X_{2,\mathbf{A}_*^{(1)}}(\mathbf{F}_*^{(1)}) \tag{35}$$

$$\|\mathbf{F}^{(2)} - \mathbf{F}_*^{(2)}\|_F \le h_2 \mathrm{Lip}(X_{2,\mathbf{F}^{(1)}})\varepsilon_2 + h_1 \mathrm{Lip}(X_{1,\mathbf{F}^{(0)}})\varepsilon_2 + \varepsilon_1. \tag{36}$$

This leads to the general bound on the expansivity of $\mathcal{D}$ when it is composed of $L$ layers, which is

$$d(\mathcal{D}(\mathbf{F}^{(0)}, \mathbf{A}^{(0)}), \mathcal{D}(\mathbf{F}_*^{(0)}, \mathbf{A}_*^{(0)})) := \| \mathrm{vec}(\mathbf{A}^{(L)}) - \mathrm{vec}(\mathbf{A}_*^{(L)})\|_1 + \|\mathbf{F}^{(L)} - \mathbf{F}_*^{(L)}\|_F$$

$$\le \varepsilon_1 + \varepsilon_2 \left( 1 + \sum_{i=1}^{L} \mathrm{Lip}(X_{i,\mathbf{F}^{(i-1)}}) h_i \right) \tag{37}$$

$$=: \varepsilon_1 + c(h_1, \ldots, h_L)\varepsilon_2.$$

We remark that in case the adjacency matrix is not perturbed, i.e. $\varepsilon_2 = 0$, the map $\mathcal{D}$ can be controlled by the perturbation magnitude to the feature matrix, i.e. $\varepsilon_1$. On the other hand, even if the features are not perturbed, i.e. $\varepsilon_1 = 0$, the feature updates can not be bounded simply with $\varepsilon_2$, since their update depends on different adjacency matrices. We have already commented on this aspect in Section 4, since this interconnection is also the reason why we have proposed to jointly update the feature and the adjacency matrices. The important aspect of this analysis is that constraining the map $\mathcal{D}$ so this contractive setup occurs allows getting the bound in Equation (37), which quantifies how sensitive the network is to perturbations.

## G  ARCHITECTURE

We describe the architecture of CSGNN in Algorithm 1.

## H  DATASETS

We provide the statistics of the datasets used in our experiments in Table 2.

---

**Algorithm 1** CSGNN Architecture

    **Input:** Attacked node features $\mathbf{F}_* \in \mathbb{R}^{n \times c_{\text{in}}}$ and adjacency matrix $\mathbf{A}_* \in \{0,1\}^{n \times n}$.
    **Output:** Predicted node labels $\tilde{\mathbf{Y}} \in \mathbb{R}^{n \times c_{\text{out}}}$.

1: **procedure** CSGNN
2:     $\mathbf{F}_* \leftarrow \text{Dropout}(\mathbf{F}_*, p)$
3:     $\mathbf{F}_*^{(0)} = \mathcal{K}(\mathbf{F}_*); \quad \mathbf{A}_*^{(0)} = \mathbf{A}_*$
4:     **for** $l = 1 \ldots L$ **do**
5:         $\mathbf{F}_*^{(l-1)} \leftarrow \text{Dropout}(\mathbf{F}_*^{(l-1)}, p)$
6:         Node Feature Dynamical System Update: $\mathbf{F}_*^{(l)} = \Psi_{X_l}^{h_l}(\mathbf{F}_*^{(l-1)}, \mathbf{A}_*^{(l-1)})$
7:         Adjacency Dynamical System Update: $\mathbf{A}_*^{(l)} = \Psi_{Y_l}^{h_l}(\mathbf{A}_*^{(l-1)})$
8:     **end for**
9:     $\mathbf{F}_*^{(L)} \leftarrow \text{Dropout}(\mathbf{F}_*^{(L)}, p)$
10:    $\tilde{\mathbf{Y}} = \mathcal{P}(\mathbf{F}_*^{(L)})$
11:    Return $\tilde{\mathbf{Y}}$
12: **end procedure**

---

Table 2: Datasets Statistics. Following (Zügner et al., 2018; Zügner & Günnemann, 2019), we consider only the largest connected component (LCC).

| Dataset | $\mathbf{N_{LCC}}$ | $\mathbf{E_{LCC}}$ | **Classes** | **Features** |
|---|---|---|---|---|
| Cora (McCallum et al., 2000) | 2,485 | 5,069 | 7 | 1,433 |
| Citeseer (Sen et al., 2008) | 2,110 | 3,668 | 6 | 3,703 |
| Polblogs (Adamic & Glance, 2005) | 1,222 | 16,714 | 2 | / |
| Pubmed (Namata et al., 2012) | 19,717 | 44,338 | 3 | 500 |

## I   HYPERPARAMETERS

All the hyperparameters were determined by grid search, and the ranges and sampling distributions are provided in Table 3.

Table 3: Hyperparameter ranges

| Hyperparameter | Range | Distribution |
|---|---|---|
| input/output embedding learning rate | $[10^{-5}, 10^{-2}]$ | uniform |
| node dynamics learning rate | $[10^{-5}, 10^{-2}]$ | uniform |
| adjacency dynamics learning rate | $[10^{-5}, 10^{-2}]$ | uniform |
| input/output embedding weight decay | $[5 \cdot 10^{-8}, 5 \cdot 10^{-2}]$ | log uniform |
| node dynamics weight decay | $[5 \cdot 10^{-8}, 5 \cdot 10^{-2}]$ | log uniform |
| adjacency dynamics weight decay | $[5 \cdot 10^{-8}, 5 \cdot 10^{-2}]$ | log uniform |
| input/output embedding dropout | $[0, 0.6]$ | uniform |
| node dynamics dropout | $[0, 0.6]$ | uniform |
| share weights between time steps | $\{\text{yes, no}\}$ | discrete uniform |
| step size $h$ | $[10^{-2}, 1]$ | log uniform |
| adjacency contractivity parameter $\alpha$ | $[-2, 0]$ | uniform |
| #layers $L$ | $\{2, 3, 4, 5\}$ | discrete uniform |
| #channels $c$ | $\{8, 16, 32, 64, 128\}$ | discrete uniform |

## J   EXPERIMENTAL RESULTS

**Results on Pubmed.** We now provide our results on the Pubmed (Namata et al., 2012) dataset, with three types of attacks: (i) non-targeted using metattack, reported in Table 4, (ii) targeted attack using nettack in Figure 5, and (iii) random adjacency matrix attack in Figure 6. Those experiments are done

under the same settings in as Section 5 in the main paper. Overall, we see that our CSGNN achieves similar or better results compared with other baselines. Specifically, we see that CSGNN outperforms all the considered baselines under targeted attack (using nettack), and similar performance when no perturbations occur, as can be depicted in Figure 5.

Table 4: Node classification performance (accuracy±std) under non-targeted attack (metattack) on the Pubmed dataset with varying perturbation rates.

| Dataset | Ptb Rate (%) | 0 | 5 | 10 | 15 | 20 | 25 |
|---------|--------------|---|---|----|----|----|----|
| Pubmed | GCN | 87.19±0.09 | 83.09±0.13 | 81.21±0.09 | 78.66±0.12 | 77.35±0.19 | 75.50±0.17 |
| | GAT | 83.73±0.40 | 78.00±0.44 | 74.93±0.38 | 71.13±0.51 | 68.21±0.96 | 65.41±0.77 |
| | RGCN | 86.16±0.18 | 81.08±0.20 | 77.51±0.27 | 73.91±0.25 | 71.18±0.31 | 67.95±0.15 |
| | GCN-Jaccard | 87.06±0.06 | 86.39±0.06 | 85.70±0.07 | 84.76±0.08 | 83.88±0.05 | 83.66±0.06 |
| | GCN-SVD | 83.44±0.21 | 83.41±0.15 | 83.27±0.21 | 83.10±0.18 | 83.01±0.22 | 82.72±0.18 |
| | Pro-GNN-fs | 87.33±0.18 | **87.25±0.09** | **87.25±0.09** | **87.20±0.09** | 87.09±0.10 | 86.71±0.09 |
| | Pro-GNN | 87.26±0.23 | 87.23±0.13 | 87.21±0.13 | 87.20±0.15 | **87.15±0.15** | **86.76±0.19** |
| | CSGNN | **87.36±0.02** | 87.16±0.10 | 87.08±0.09 | 87.06±0.08 | 86.59±0.18 | 86.63±0.08 |

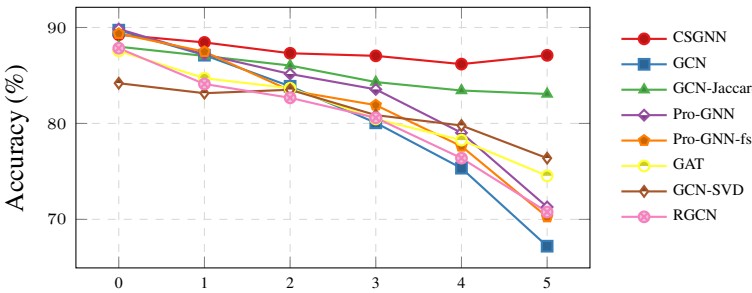

Figure 5: Node classification accuracy (%) on the Pubmed dataset using nettack as attack method. The horizontal axis describes the number of perturbations per node.

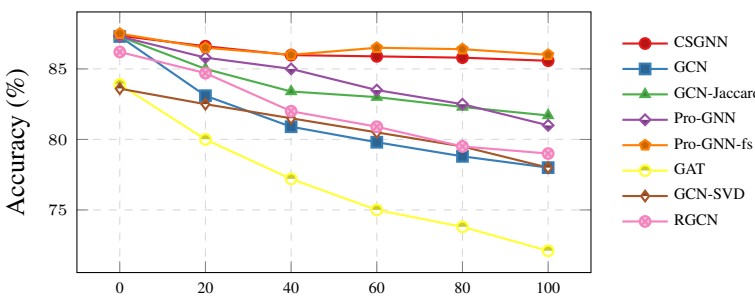

Figure 6: Node classification accuracy (%) on the Pubmed dataset with a random adjacency matrix attack. The horizontal axis describes the attack percentage.

**Absolute performance results on Adaptive Attacks.** In Section 5.2 we provide the obtained relative node classification accuracy (%) with respect to the accuracy of GCN. Here, we also provide the absolute results, for an additional perspective on the performance of CSGNN. Our results are reported in Figure 7.

**Enforcing contractive node dynamics improves baseline performance.** As discussed in Section 4.2, we draw inspiration from contractive dynamical systems, and therefore propose a contractivity-inspired node feature dynamical system. For our results in the main paper, in Section 5, we use a more general definition of the node dynamical system, that can admit both contractive and non-contractive

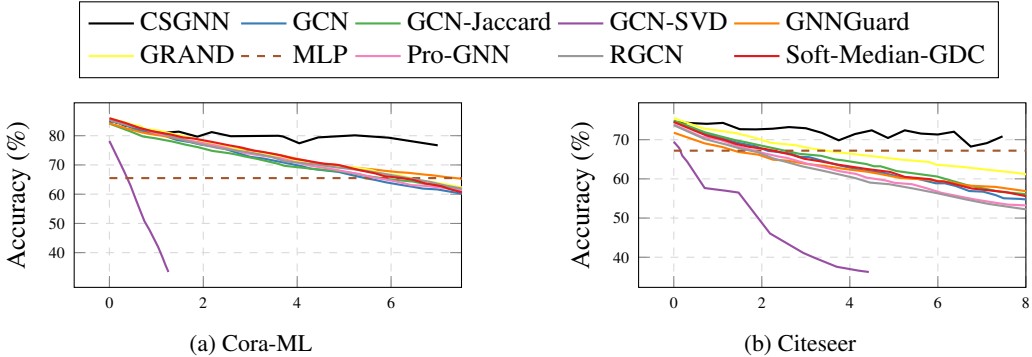

Figure 7: Absolute results version of Figure 4. The horizontal axis describes the attack budget (%) as defined in Mujkanovic et al. (2022).

Table 5: Node classification performance (accuracy±std) of ECSGNN and other baselines, under non-targeted attack (metattack) with varying perturbation rates.

| Dataset | Ptb Rate (%) | 0 | 5 | 10 | 15 | 20 | 25 |
|---------|--------------|---|---|----|----|----|----|
| Cora | GCN | 83.50±0.44 | 76.55±0.79 | 70.39±1.28 | 65.10±0.71 | 59.56±2.72 | 47.53±1.96 |
| | GAT | 83.97±0.65 | 80.44±0.74 | 75.61±0.59 | 69.78±1.28 | 59.94±0.92 | 54.78±0.74 |
| | RGCN | 83.09±0.44 | 77.42±0.39 | 72.22±0.38 | 66.82±0.39 | 59.27±0.37 | 50.51±0.78 |
| | GCN-Jaccard | 82.05±0.51 | 79.13±0.59 | 75.16±0.76 | 71.03±0.64 | 65.71±0.89 | 60.82±1.08 |
| | GCN-SVD | 80.63±0.45 | 78.39±0.54 | 71.47±0.83 | 66.69±1.18 | 58.94±1.13 | 52.06±1.19 |
| | Pro-GNN-fs | 83.42±0.52 | 82.78±0.39 | 77.91±0.86 | 76.01±1.12 | 68.78±5.84 | 56.54±2.58 |
| | Pro-GNN | 82.98±0.23 | 82.27±0.45 | 79.03±0.59 | 76.40±1.27 | 73.32±1.56 | 69.72±1.69 |
| | Mid-GCN | 84.61±0.46 | 82.94±0.59 | 80.14±0.86 | 77.77±0.75 | 76.58±0.29 | 72.89±0.81 |
| | CSGNN | 84.12±0.31 | 82.20±0.65 | 80.43±0.74 | 79.32±1.04 | 77.47±1.22 | 74.46±0.99 |
| | ECSGNN | 82.79±0.33 | 80.59±0.61 | 79.19±0.81 | 76.29±0.96 | 73.88±0.84 | 72.27±0.78 |
| Citeseer | GCN | 71.96±0.55 | 70.88±0.62 | 67.55±0.89 | 64.52±1.11 | 62.03±3.49 | 56.94±2.09 |
| | GAT | 73.26±0.83 | 72.89±0.83 | 70.63±0.48 | 69.02±1.09 | 61.04±1.52 | 61.85±1.12 |
| | RGCN | 71.20±0.83 | 70.50±0.43 | 67.71±0.30 | 65.69±0.37 | 62.49±1.22 | 55.35±0.66 |
| | GCN-Jaccard | 72.10±0.63 | 70.51±0.97 | 69.54±0.56 | 65.95±0.94 | 59.30±1.40 | 59.89±1.47 |
| | GCN-SVD | 70.65±0.32 | 68.84±0.72 | 68.87±0.62 | 63.26±0.96 | 58.55±1.09 | 57.18±1.87 |
| | Pro-GNN-fs | 73.26±0.38 | 73.09±0.34 | 72.43±0.52 | 70.82±0.87 | 66.19±2.38 | 66.40±2.57 |
| | Pro-GNN | 73.28±0.69 | 72.93±0.57 | 72.51±0.75 | 72.03±1.11 | 70.02±2.28 | 68.95±2.78 |
| | Mid-GCN | 74.17±0.28 | 74.31±0.42 | 73.59±0.29 | 73.69±0.29 | 71.51±0.83 | 69.12±0.72 |
| | CSGNN | 74.93±0.52 | 74.91±0.33 | 73.95±0.35 | 73.82±0.61 | 73.01±0.77 | 72.94±0.56 |
| | ECSGNN | 75.01±0.28 | 74.97±0.38 | 73.97±0.29 | 73.67±0.45 | 72.92±0.97 | 72.89±0.90 |

dynamics, in a data-driven fashion that generalizes the contractive behavior described in Theorem 2. To motivate our choice and inspiration from such systems, we now show that by enforcing contractive dynamics only (i.e., ensuring $\tilde{\mathbf{K}}_l$ is positive definite), improved results are also achieved, in addition to our results with CSGNN in the main paper. To this end, we will denote the *Enforced* CSGNN variant of our method by ECSGNN. We present the performance of ECSGNN under the non-targeted metattack in Table 5, targeted nettack in Figure 8, and random attacks in Figure 9. Overall, we see that ECSGNN typically offers improved performance compared to several baselines, and in some cases outperforms all of the considered models. Also, we find that its extension, non-enforced CSGNN tends to yield further performance improvements, as shown in the main paper. Our conclusion from this experiment is that node feature contractivity helps to improve robustness to adversarial attacks.

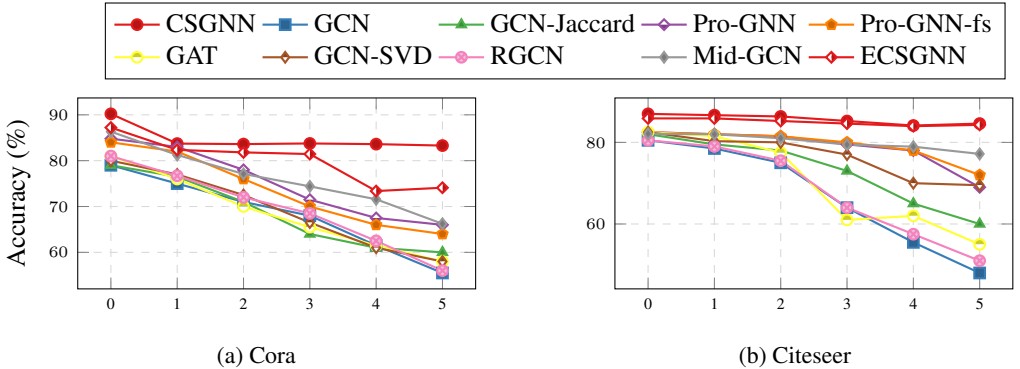

Figure 8: Node classification accuracy (%) of ECSGNN and other baselines, under a targeted attack generated by nettack. The horizontal axis describes the number of perturbations per node.

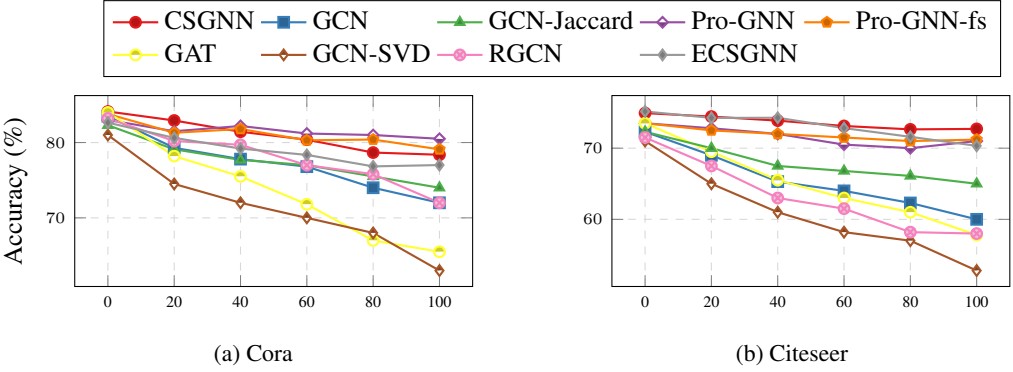

Figure 9: Node classification accuracy (%) of ECSGNN and other baselines, under a random adjacency matrix attack. The horizontal axis describes the attack percentage.

**Learning contractive adjacency dynamics is beneficial.** As our CSGNN is composed of two learnable coupled dynamical systems that evolve both the node features and the adjacency matrix, it is interesting to quantify the importance of learning the adjacency matrix dynamics. Therefore, we now report the node classification accuracy (%) on the Cora and Citeseer datasets, under three different attack settings - non-targeted with metattack, targeted with nettack, and a random adjacency matrix attack. We choose the strongest attack under each setting, out of our experiments in Section 5.2. Namely, for metattack, we choose the highest perturbation rate of 25%, for nettack we choose the maximal number of node perturbations of 5, and for random adjacency matrix attack we randomly add edges that amount to 100% of the edges in the original, clean graph. We report the results in Table 6. We denote the CSGNN variant that does not employ an adjacency dynamical system by CSGNN$_{noAdj}$. As can be seen from the table, there is a positive impact when adding the learnable adjacency matrix component to our CSGNN, highlighting its importance to the learned dynamical system.

| Method | Cora | | | Citeseer | | |
|---|---|---|---|---|---|---|
| | nettack | metattack | random | nettack | metattack | random |
| CSGNN$_{noAdj}$ | 81.90 | 70.25 | 77.19 | 82.20 | 70.17 | 71.28 |
| CSGNN | 83.29 | 74.46 | 78.38 | 84.60 | 72.94 | 72.70 |

Table 6: The influence of learning the adjacency dynamical system $\Psi_{Y_t}^{h_l}$. The results show the node classification accuracy (%) with and without learning the adjacency dynamical system.

**Comparison with GNNGuard.** We now provide a comparison of our CSGNN with GNNGuard Zhang & Zitnik (2020). We report results, both on Cora and Citeseer, for Metattack with a 20% perturbation rate and for 5 targeted nodes using Nettack, as reported in Zhang & Zitnik (2020). These results further highlight the significance of our method, given that in most cases, CSGNN outperforms GNNGuard.

| Method | Cora | | Citeseer | |
|---|---|---|---|---|
| | metattack, 20% | nettack, 5 targets | metattack, 20% | nettack, 5 targets |
| GNNGuard | 72.20 | 77.50 | 71.10 | 86.50 |
| CSGNN (Ours) | 77.47 | 83.20 | 73.00 | 84.60 |

Table 7: Comparison of our proposed GNN architecture with GNNGuard, based on the experimental setup proposed in Zhang & Zitnik (2020). The results show the node classification accuracy (%).

## K  COMPLEXITY AND RUNTIMES

This appendix provides an analysis of the complexity and runtimes of our proposed graph neural network, accounting for the additional overhead cost due to the updates of the adjacency matrix. Inference times and memory consumption are based on experiments run on the Cora dataset with an Nvidia RTX-3090 GPU with 24GB of memory. In theory, the added runtime complexity for a CSGNN adjacency matrix learning layer is of $O(9 \cdot n^2)$ where $n$ is the number of nodes. Thus, assuming $L$ layers, the overall complexity of a CSGNN network is $O(L(n \cdot c^2 + m \cdot c + 9 \cdot n^2))$, whereas node-based ODE systems usually are of runtime complexity $O(L(n \cdot c^2 + m \cdot c))$. We recall that $n$ is the number of nodes, $m$ is the number of edges in the graph, and the term of $c^2$ stems from the channel mixing term in Equation (8). The factor of 9 stems from the 9 parameters to be learned in Equation Equation (14). Below, we report the runtimes and memory consumption of our CSGNN and compare it with CSGNN$_{noAdj}$ (that is, CSGNN without the adjacency matrix update), and Pro-GNN for reference. These experiments were run on the Cora dataset with networks with 64 channels and 2 layers. It can be seen that both CSGNN and Pro-GNN require more resources, however, such an approach offers improved performance.

Also, we note that while both our CSGNN and Pro-GNN propose methods to evolve the adjacency matrix, our CSGNN shows significantly lower training computational time, due to our solution taking the form of a learned neural ODE system for the adjacency matrix, while Pro-GNN solves an

| Method | Time (ms) | | Memory (MB) | Classification accuracy (%) | |
|---|---|---|---|---|---|
| | Training | Inference | | metattack, 25% | nettack, 5 targets |
| Pro-GNN | 1681.17 | 1.01 | 1989 | 69.72 | 66.21 |
| CSGNN$_{noAdj}$ | 3.29 | 1.32 | 891 | 70.25 | 81.90 |
| CSGNN | 9.24 | 5.96 | 1623 | 74.46 | 83.29 |

Table 8: Comparison of our proposed GNN architecture with Pro-GNN in terms of runtime in milliseconds (measured per epoch) and memory consumption in megabytes, as well as the accuracy (%) of the models under different attacks. The cost of the training refers to one epoch.

optimization problem with feedback to the downstream task (which is also an end-to-end solution, although different than ours). For inference, Pro-GNN uses a fixed adjacency matrix that is found by training, while our CSGNN evolves the input adjacency matrix using the learned network. This can also be seen as an advantage of CSGNN, as it can be used for inference on different kinds of attacks, and therefore can potentially generalize to different attacks, and this is a future research direction.

To conclude, we remark that investigation of techniques that allow for a reduction of the computational costs in the updates of the adjacency matrices is an important and interesting topic on its own, that is left for future work.

