# OpenReview forum: "Contractive Systems Improve Graph Neural Networks Against Adversarial Attacks"
_ICLR.cc/2024/Conference — Submitted to ICLR 2024_

### Official Review · Reviewer_mRqu · 2023-10-20

**Soundness:** 3 good
**Presentation:** 2 fair
**Contribution:** 2 fair
**Rating:** 5
**Confidence:** 4

**Summary:**

A neural diffusion GNN model coupling the evolution of both node features and graph adjacency matrix is proposed. Analytical studies on the contractive properties of this model across model layers are provided.

**Strengths:**

Using a diffusion process to model the joint evolution of node features and graph adjacency matrix seems to be novel. Numerical experiments indicate that such an approach can provide robustness against adversarial attacks.

**Weaknesses:**

1. The analytical results show contractive properties of the feature/adjacency matrix evolution across model layers for a given input. I am unclear how this proves robustness of the GNN model to input or structure perturbation.

1. Missing comparison to the work “On the robustness of graph neural diffusion to topology perturbations,” NeurIPS 2022. What are the additional things we learn from this current paper? The results in the NeurIPS 2022 paper relate explicitly to robustness w.r.t. input perturbations.

**Questions:**

1. Please give more information on the attack type. Is it inductive, modification/injection, whitebox/blackbox?

1. What is the adaptive attack procedure? It seems the paper simply uses unit tests from Mujkanovic et al. (2022). These cannot be considered to be adaptive attacks for the proposed model. Mujkanovic et al. (2022) has emphasized this point too: "we cannot stress enough that this collection does not replace a properly developed adaptive attack".

1. GNNGuard is mentioned but not used as baseline in Table 1.

1. The attacks used do not seem strong enough (I am unclear of their settings as well). E.g., in Table 1, even under 25% attack, GCN still has >40% accuracy. In other related papers on GNN adversarial attacks (e.g., Fig. 1 of Mujkanovic et al. (2022)), usually GCN would have performed much worse with accuracies below 20-30%.

---

> ### Author Response · Authors · 2023-11-16
> **Response (Part 1)**
>
> We sincerely appreciate the review comments regarding the novelty of our work and its performance, stating that ‘Using a diffusion process to model the joint evolution of node features and graph adjacency matrix seems to be novel.’ and that it provides ‘robustness against adversarial attacks’. We thank the reviewer for the insightful comments and questions, which we address below. In our revised paper, our responses to your queries are marked in **Orange**.
>
>
> **Regarding why contractivity improves robustness**: As defined in our original submission (please see Equations (10) and (11)), contractivity leads to the reduction of the sensitivity of network outputs to input perturbations. Because graph attacks are based on input perturbations such as changes to the node features or adjacency matrix, and our CSGNN encapsulates an inductive bias in the form of a contractive system, it learns to fortify itself against graph attacks (i.e., perturbations), thereby improving the robustness of the model. More explicitly, even without the knowledge of the clean adjacency and feature matrices, having a contractive and stable system allows the outputs one gets with perturbed inputs to be close to the one would get feeding in the clean inputs. Indeed, this understanding is also empirically validated throughout our extensive experiments in Section 4. We discuss this point in our Preliminaries section (Section 3), and have now added an appendix section (Appendix A), to provide a better background on contractive systems.
>
> **Regarding  "On the robustness of graph neural diffusion to topology perturbations" (NeurIPS 2022)**: We thank you for this important reference, which we now cite and discuss in our revised paper. The most crucial difference is the underlying method, novelty, and the performance of the methods on various benchmarks. While the mentioned paper also takes inspiration from neural diffusion GNNs, it utilizes **only node feature update neural ODEs**. On the contrary, our CSGNN offers a novel **coupled neural ODE that learns both node features and adjacency matrix updates**. This addition is proven to provide improved performance, as reported in the paragraph “Learning contractive adjacency dynamics is beneficial” of Appendix J (of the revised version of the manuscript). As 3 of 4 of the reviewers noted, this is a novel approach.
>
> **Regarding attack type:** We follow the experimental settings in Pro-GNN, which considers inductive poisoning attacks using both white (metattack, nettack) and black box (random) configurations. In our original submission, we mentioned that in our experimental section, in Section 5.2. We have also revised our Introduction to highlight that, following your question.
>
> **Regarding adaptive attack**: The reviewer is correct that we use the ‘unit-tests’ from Mujkanovic et al. (2022)., as stated in our original submission. As we discuss in Section 5.2, under the paragraph “Robustness to Adaptive Attacks”, the unit-tests are adaptive in that a different attack can be utilized for a different attack strength. We also note regarding the statement in Mujkanovic et al. (2022), “we cannot stress enough that this collection does not replace a properly developed adaptive attack”, that our goal is not to propose new attacks based on our method but rather to design a defensive mechanism. Thus, our motivation for using those useful unit tests was to be able to compare with recent baselines on challenging test cases, as shown in Figures 4 and 7. To address your concern, we have revised our text to reflect this discussion. Please see our changes in the revised paper, marked in Orange.

---

> > ### Comment · Reviewer_mRqu · 2023-11-17
> >
> > Thank you for the clarifications.
> >
> > Perhaps, I was not specific enough in my initial comment. In eq. (10) of Theorem 2, $\mathbf{A}$ is fixed but clearly $\Psi_{X_l}^{h_l}$ is impacted by $\mathbf{A}*$. You cannot apply eq. (11) here as $\Psi_{X_l}^{h_l}$ and $\Psi_{Y_l}^{h_l}$ are two different functions. Therefore, I am unclear how the **individual** contractivity of feature/adjacency matrix evolution imply robustness.
> >
> > Since unit tests from Mujkanovic et al. (2022) do not imply robustness against adaptive attacks for this model, I don't think the authors should continue to claim that in the paper. This is misleading for the readers.
> >
> > There are many other adversarial attacks like injection attacks. This paper only tests on modification attacks using Metattack and NETTACK. If the paper's focus is only on modification attacks, one can argue that the paper title and abstract do not reflect the technical content. Moreover, what is the motivation to focus only on modification attacks? Are these more important in practice?

---

> > > ### Author Response · Authors · 2023-11-17
> > > **Response by authors**
> > >
> > > We thank the Reviewer for the quick response. We sincerely appreciate your engagement! Please find our responses to your queries below.
> > >
> > > **Regarding the contractivity of the composition of the Explicit Euler steps**: The reviewer is correct. Indeed, as we comment in  Appendix D and F of the updated manuscript, the composition of the two systems is not necessarily contractive. However, the dynamics for the adjacency matrix is uncoupled from that of the features, and it is contractive. This part of the network has been our primary focus since this contractive behavior makes the updates to the adjacency matrix stable to perturbations. On the other hand, having contractive feature dynamics allows for a bound of the Lipschitz constant of the overall network, which couldn't be that tight if it were not contractive. There is a very well-developed research direction based on Lipschitz-constrained neural networks to improve the robustness. Making networks 1-Lipschitz, i.e., contractive, is not the general goal in these works. Having networks being 1-Lipschitz is convenient because one can obtain an arbitrary Lipschitz constant by suitable rescalings and exploit this property to improve the robustness or get robustness certificates. Our aim with this paper is to provide a theoretical analysis of this coupled system, focusing on what can be done on the design of the individual systems to make their interaction well-behaved, i.e. with a controllable Lipschitz constant. In the experiments, we see that even without forcing this Lipschitz constant to the (arbitrary) value of 1, the networks obtained in a data-driven manner are more robust than the other models we compare with and which have been designed with a similar purpose as ours.
> > >
> > > We agree that it could be interesting to consider other coupling strategies and see if better bounds on the expansivity of the network can be obtained. However, the experimental results in the manuscript are strong evidence of the benefit of employing the proposed architectural choice and coupled dynamical system.
> > >
> > > **Regarding the attacks from Mujkanovic et al. (2022)**: We thank the reviewer for the comment. Clearly, our intention is not to mislead the readers, but rather to give a principled name to describe the attacks considered in the experiments shown in Figures 4 and 7. We will change the name of the attacks from 'adaptive' to 'additional' unit tests, to better distinguish between tests that are adaptive to the model and the unit tests in Mujkanovic et al. (2022).
> > >
> > >
> > > **Regarding injection attacks**: As stated in our original submission, and further highlighted in our revised paper, we focus on poisoning attacks. The reason for that is that it reflects a more realistic attack than, say, node feature noising attacks, as discussed in Pro-GNN. Furthermore, as that kind of attacks were demonstrated in many existing methods, we intended to be able to compare with as many methods as possible. It is also important to note that our CSGNN will continue to work even under a change in the number of nodes (translating to injection attacks) because the parameterization of our adjacency matrix ODE system is agnostic to the number of nodes in the graph, being it dependent only on 9 scalar values. This is another advantage of our method.
> > >
> > > We agree that the case of injection attacks is also a very realistic scenario, and to address your request, we are now experimenting with such attacks as well. To provide reliable and profound results, including an extension of our theoretical section to such cases, we will appreciate the patience of the Reviewer that will allow us to add those results.
> > >
> > >
> > > In the meantime, we are happy to discuss any other standing doubts or concerns you may have.

---

> > > ### Author Response · Authors · 2023-11-19
> > > **Experimenting with injection attacks**
> > >
> > > Dear Reviewer mRqu,
> > >
> > > We have tried to experiment with the popular injection attacks offered by the python package 'grb' (please see [S1] for the paper below). These attacks are also the ones used in the paper proposed by the Reviewer,  “On the robustness of graph neural diffusion to topology perturbations,” NeurIPS 2022. , and because they also rely on a ODE perspective of GNNs (although considering only the node features), it will allow us to directly compare with them and many other methods, as this is a popular injection attack benchmark. **However, the attacks and data are not available.**
> > >
> > > We have contacted the authors asking to receive assistance with gaining access to the attacks and data so we can add the results during the rebuttal phase. In any case, we will add the results as soon as the data becomes available.
> > >
> > > We are also happy to experiment with other injection benchmarks, should the Reviewer have alternative suggestions.
> > >
> > > We thank you for your engagement,
> > >
> > > The authors.
> > >
> > > [S1] Graph Robustness Benchmark: Benchmarking the Adversarial Robustness of Graph Machine Learning

---

> > > > ### Author Response · Authors · 2023-11-21
> > > > **A message to reviewer mRqu**
> > > >
> > > > Dear Reviewer mRqu,
> > > >
> > > > We would like to update you that so far we have no received any response regarding the data, however as promised we will add the results as soon as we have access to the data.
> > > >
> > > > We believe that our rebuttal and responses to your review and other reviews has a larger scope than only those additional (and important) experiments, and therefore we would love to know if you're satisfied with our overall rebuttal. In a positive case, we would like to gently ask you to consider to revise your score.
> > > >
> > > > Best regards,
> > > >
> > > > The authors.

---

> > > > > ### Comment · Reviewer_mRqu · 2023-11-22
> > > > >
> > > > > Thank you for all the clarifications. While the claimed novelty of this paper is the coupled feature and adjacency matrix update model, the result only shows that updates to the adjacency matrix are stable to perturbations. More work needs to be done. I am therefore keeping my score.

---

> > > > > > ### Author Response · Authors · 2023-11-22
> > > > > >
> > > > > > We thank the reviewer for the answer. Let us clarify here what is provided in the paper regarding the stability of the joint dynamics. While it is true that the adjacency matrix updates are guaranteed to be contractive on their own, we do not agree that this is the only result provided in the paper. We also provide a theoretical analysis of the expansivity of the whole network. More precisely, in Appendix F, we compute a bound on the Lipschitz constant of the layers given by explicit Euler steps, i.e. the map $\mathcal{D}$. The Lipschitz constant of $\mathcal{D}$ can in principle forced to be close to 1, for example by penalizing larger steps in the loss function. We remark in the manuscript that our purpose was to experiment with a fully data-driven adaptation of the expansivity of the layers, with the aim of maximizing the robust accuracy of the network. Furthermore, in the appendices we show that imposing the contractivity in the feature updates provides improved or on-par results with respect to the other methods we compared with.

---

> ### Author Response · Authors · 2023-11-16
> **Response (Part 2)**
>
> **Regarding GNNGuard in Table 1**: In our experimental results section, we seek to provide a comprehensive comparison with the latest methods. We did not include GNNGuard in Table 1 because there were no reported results on Metattack using a variable % of attack in the original paper. However, to address the reviewer's question, we provide a comparison below with GNNGuard on Metattack with a 20% perturbation rate and 5 targeted nodes using Nettack, as reported in GNNGuard, both on Cora and on Citeseer.
>
> | Cora |              |                                             |                                           |
> |------|--------------|---------------------------------------------|-------------------------------------------|
> |      | Method       | Accuracy on Metattack 20% perturbation rate | Accuracy on Nettack with 5 targeted nodes |
> |      | GNNGuard     | 72.2                                        | 77.5                                      |
> |      | CSGNN (Ours) | 77.4                                   | 83.2                                  |
>
>
> | Citeseer |              |                                             |                                           |
> |----------|--------------|---------------------------------------------|-------------------------------------------|
> |          | Method       | Accuracy on Metattack 20% perturbation rate | Accuracy on Nettack with 5 targeted nodes |
> |          | GNNGuard     | 71.1                                        | 86.5                                      |
> |          | CSGNN (Ours) | 73.0                                        | 84.6                                      |
>
>
> Those results further highlight the significance of our method, given that in most cases, CSGNN outperforms GNNGuard. We have added those results to our revised paper marked in Orange, in the paragraph titled “Comparison with GNNGuard” of Appendix J.
>
> **Regarding the strength of the attack**: In our experimental section, we provide results using various settings that follow those presented in Pro-GNN and Mujkanovic et al. (2022). This approach allows us to provide an accurate and comprehensive picture of the current state of GNN defense mechanisms. Furthermore, we would like to kindly note that we include recent tests from  Mujkanovic et al. (2022), that indeed seem to be stronger. Still, even under this harder attack, our CSGNN achieves improved results compared to other methods, as can be seen in Figures 4 and 7.

---

### Official Review · Reviewer_SPJa · 2023-10-31

**Soundness:** 3 good
**Presentation:** 3 good
**Contribution:** 3 good
**Rating:** 5
**Confidence:** 4

**Summary:**

The authors introduce an approach to enhance the robustness of GNNs against adversarial perturbations by leveraging the concept of contractive dynamical systems.

**Strengths:**

1. The paper presents a novel architecture, CSGNN, that innovatively integrates the principles of contractive dynamical systems to enhance the robustness of GNNs against adversarial poisoning attacks.
2. The simultaneous evolution of node features and the adjacency matrix is a distinctive feature that can potentially offer intrinsic resistance to adversarial perturbations.
3. The authors fortify their claims with a rigorous theoretical analysis.

**Weaknesses:**

1. Inadequate Literature Review: The paper's glaring omission of pivotal related works is concerning. The idea of enhancing NN robustness via dynamical systems isn't novel, even within the GNN realm. The authors' failure to acknowledge, let alone differentiate their work from seminal papers [1][2][3][4][5], is a significant oversight.


2. Lack of Clear Motivation: The paper's design choices seem arbitrary, with many equations appearing devoid of clear rationale. For example:
>The reasoning behind assuming a piecewise constant function in eq(6).
>The ambiguity surrounding the gradient operator of $A$ in eq(8).
>The seemingly ad-hoc design of eq(8) and its alignment with the paper's theorems.
>The choice to enforce symmetry on $\tilde{\mathbf{K}}_l$.
>The intricate design of the adjacency matrix update in eq(14) lacks clear justification.
The paper should not be a mere mathematical exercise; it should be accessible and provide clear motivations for design choices.

3. Reproducibility Concerns: The absence of code hinders the verification of the paper's claims. Critical aspects, such as adherence to the adjacency update mechanism in eq(14) and the positive definiteness of $\tilde{\mathbf{K}}_l$, remain unchecked.

4. Computational Overheads: The complete matrix representation in eq(14) suggests significant computational demands. The authors should elucidate the memory and time overheads.

5. Narrow Attack Scope:
The paper exclusively focuses on poisoning attacks. Is this indicative of the theorems being specifically tailored for such attacks? The theorem statements, including their assumptions and conclusions, don't seem to impose such constraints. What is the rationale behind primarily considering poisoning attacks? Reference [6] suggests that injection attacks pose a greater threat than poisoning attacks. The authors should address these concerns and expand their experiments to include injection attacks, irrespective of the model's performance against them.
Additionally, the inclusion of black-box attacks in the evaluation is necessary.
The model exhibits suboptimal performance on the Pubmed dataset. Does this suggest that the efficacy of your models and theorems is dataset-dependent? It would be insightful to understand how the model fares on different datasets, especially those characterized by heterophily.

6. Lack of Large-Scale Graph Datasets:
The current evaluation is limited to smaller datasets such as Cora, Citesser, and Polblogs. It would be beneficial to see how the model performs on more extensive, widely-recognized datasets like the ogbn series.


Overall, I believe this paper has promise. However, in its present state, I cannot endorse its acceptance. I strongly urge the authors to undertake a thorough revision and consider resubmitting to an upcoming top-tier machine learning conference.

[1] Zakwan, Muhammad, Liang Xu, and Giancarlo Ferrari-Trecate. "Robust classification using contractive Hamiltonian neural ODEs." IEEE Control Systems Letters 7 (2022): 145-150.

[2] Kang, Qiyu, Yang Song, Qinxu Ding, and Wee Peng Tay. "Stable neural ode with lyapunov-stable equilibrium points for defending against adversarial attacks." Advances in Neural Information Processing Systems 34 (2021): 14925-14937.

[3] Huang, Yujia, Ivan Dario Jimenez Rodriguez, Huan Zhang, Yuanyuan Shi, and Yisong Yue. "Fi-ode: Certified and robust forward invariance in neural odes." arXiv preprint arXiv:2210.16940 (2022).

[4] Yang, Runing, Ruoxi Jia, Xiangyu Zhang, and Ming Jin. "Certifiably Robust Neural ODE with Learning-based Barrier Function." IEEE Control Systems Letters (2023).

[5] Song, Yang, Qiyu Kang, Sijie Wang, Kai Zhao, and Wee Peng Tay. "On the robustness of graph neural diffusion to topology perturbations." Advances in Neural Information Processing Systems 35 (2022): 6384-6396.

[6] Chen, Yongqiang, Han Yang, Yonggang Zhang, Kaili Ma, Tongliang Liu, Bo Han, and James Cheng. "Understanding and improving graph injection attack by promoting unnoticeability." arXiv preprint arXiv:2202.08057 (2022).

**Questions:**

NA

---

> ### Author Response · Authors · 2023-11-16
> **Response (Part 1)**
>
> We thank the reviewer for the thorough and detailed feedback. The reviewer says that our paper proposes  ‘a novel architecture, CSGNN, that innovatively integrates the principles of contractive dynamical systems to enhance the robustness of GNNs against adversarial poisoning attacks’ and that ‘The authors fortify their claims with a rigorous theoretical analysis.’ We thank you for acknowledging the novelty of our work.  In our revised paper, our responses to your queries are marked in **Magenta**.
>
>
>
>
> The reviewer also suggests several important points to improve our paper. We now address them individually. Our changes to the submission are marked in Magenta. We hope that the reviewer is satisfied with our updated paper and responses below and that they will consider raising their score.  We are also happy to continue the discussion and address any remaining questions.
>
>
> **Regarding literature review:** We thank the reviewer for the important references, which we now include and discuss in our revised version. Specifically, we note that papers [1-4] consider structured grid data (e.g., 2D images), while our work considers graphs and GNNs. Still, we agree it is essential to discuss the papers working on adversarial robustness and following a similar approach to the one we suggest. We have now updated the manuscript, including Appendix E, where the mentioned references and a few more are included and briefly discussed. This appendix is also referenced in the Introduction section of the main body.  Regarding paper [5], it studies the effectiveness of diffusion neural GNNs to graph topology attacks, showing its resilience compared to other methods. This paper is indeed very important and also adds to the motivation for using such networks for defense. Also, please see, regarding your query about the design of Equation (8), that paper [5] proposes a similar definition of the node feature system. Regarding paper [6], it offers a way to generate new injection attacks, while our paper focuses on a defense mechanism for GNNs. We added papers [5,6] to our Related Work section and a discussion of [1-4] to our revised Appendix. Please see our changes in Magenta.
>
>
> Finally, we would like to mention that the main novelty of our work, which is the learning of a coupled dynamical system that includes the adjacency matrix, is indeed an original contribution of our work, which is not considered in [1-6]. We add to our revised paper Appendix E, that discusses papers [1-4], and include references [5-6] in the “Related Work” section of the main text.

---

> ### Author Response · Authors · 2023-11-16
> **Response (Part 2)**
>
> **Regarding motivation**: We thank the reviewer for the questions. We split our response into different parts according to the questions raised by the reviewer.
>
>
> (i) The assumption of a piecewise constant function in Equation (6) is a common practice in neural ODEs, and it allows us to assume that we can interpret the learned neural network layers as the discretization of some continuous ODE. This is a very basic assumption, and it is also made in [1] as suggested by the reviewer. Furthermore, we kindly refer the reviewer to the following papers [R1,R2,R3,R4,R5] (cited in our original submission), where a similar assumption is made. To clarify this point, we have revised our text (please see changes in Magenta).
>
>
> (ii) We would like to kindly refer the reviewer to the text above Equation (8), where we discuss the motivation for constructing our node dynamics. As stated, “We build upon a diffusion-based GNN layer, (Chamberlain et al., 2021; Eliasof et al.,2021), that is known to be stable, and under certain assumptions is contractive”. That is, the construction of this mechanism allows us to theoretically analyze and explain the behavior of the node feature dynamical system of CSGNN. Using the gradient operator in Equation (8), we rely on the definition made in (Chamberlain et al., 2021; Eliasof et al.,2021). The same definitions are also used in the paper provided by the reviewer (paper [5]). To better accommodate the reviewer’s question, we added an explicit definition of the gradient operator in our revised paper in Appendix B (which used to be Appendix A). We refer to appendix B before equation (8) in our revised version.
>
>
> (iii) Our choice to keep $\tilde{\mathbf K}_l$ symmetric is not arbitrary. It stems from [R6] (also cited in our original submission), allowing us to view the layers of the GNN as a discretization of a gradient flow. Therefore, it is important so as not to compromise the dynamical system view in our CSGNN. We have revised our paper to include this discussion.
>
>
> In addition to the findings in [R6], in our case, it is reasonable to keep $\tilde{\mathbf K}_l$ symmetric so that if it is also learned to be positive definite, then the forward propagation of our node feature system is stable, as pointed out in Appendix A (now Appendix B after the updates introduced in the revised manuscript). Our original submission also provided this discussion; please see the paragraph after Equation (10) up to Section 4.3.
>
>
>
>
> (iv) We would like to note that the design of Equation (14) is far from an arbitrary choice. As discussed in our original submission, we build on the results shown in [R7], to design an equivariant and symmetry-preserving parameterization of the adjacency matrix, which is used to define our adjacency matrix update step. Furthermore, the design of Equation (14) is a major contribution of our work, and it addresses both mathematical and technical challenges in implementing an adjacency matrix dynamical system. We gently ask the reviewer to read our discussion that motivates this structure in Section 4.3.
>
>
> **Regarding reproducibility**: We are currently not at liberty to share the source code. However, we pledge to publicly share it on GitHub upon acceptance. Additionally, we note that for reproducibility purposes, our original submission includes a detailed description of our method and architectures in Appendices G, H, and I (which used to be E, F, and G in our original submission).
>
> **Regarding checking $\mathbf K_l$ being positive-definite**: In our submission, we clearly state in Appendix B (used to be Appendix A) that depending on the weights of $K$, we may or may not get a stable update for the node features (which translates to $\tilde{\mathbf K}_l$ being positive-definite). Furthermore, **our experiments check the influence of $\tilde{\mathbf K}_l$ being positive-definite or not**. Specifically, in Appendix J, under the paragraph “Enforcing contractive node dynamics improves baseline performance”, we experiment with a variant of CSGNN that enforces the stability of the node dynamical system, called ECSGNN. Our results indicate improved performance compared to existing methods with ECSGNN, but further improvements can be obtained if we let $K$ be learned in a data-driven fashion. Our original submission included a reference to the results provided in Appendix J in the paragraph after Equation (10) up to Section 4.3. Therefore, we believe that our original submission offers a fair exposition of our work, both quantitatively and qualitatively, and we would like to hear the reviewer’s opinion on this point after our clarification. To further address your concern, we have revised Appendix J text to reflect the discussion here better.

---

> ### Author Response · Authors · 2023-11-16
> **Response (Part 3)**
>
> **Regarding computational overhead:** The reviewer is correct that learning to modify the adjacency matrix in an end-to-end fashion incurs additional computational overhead. We would like to start by noting that such an approach is not computationally light, regardless of the method (whether our CSGNN or other well-known methods like Pro-GNN [R8]). While such mechanisms do require more computations, they also offer improved performance.
> We now provide a complexity analysis of our CSGNN, followed by a table that reports the training and inference times and memory consumption on an Nvidia RTX-3090 GPU with 24GB of memory, on the Cora dataset.
> Theoretically, the added runtime complexity for a CSGNN adjacency matrix learning layer is $O(9 \cdot n^2)$ where $n$ is the number of nodes. Thus, assuming L layers, the overall complexity of a CSGNN network is $O(L(n \cdot c^2 + m\cdot c + 9\cdot n^2))$, whereas node-based ODE systems usually are of runtime complexity $O(L(n \cdot c^2 + m\cdot c))$. Please recall that $n$ is the number of nodes, $m$ is the number of edges in the graph, and the term of $c^2$ stems from the channel mixing term in Equation (8). The factor of 9 stems from the 9 parameters to be learned in Equation (14).
> Below, we report the runtimes and memory consumption of our CSGNN and compare it with CSGNN$_{\textrm{noAdj}}$ (that is, CSGNN without the adjacency matrix update), and Pro-GNN for reference. These experiments were run on the Cora dataset with networks with 64 channels and 2 layers. It can be seen that both CSGNN and Pro-GNN require more resources, however, such an approach offers improved performance.
>
>
>
>
> | Method          | Train time [ms] | Inference time [ms] | Memory consumption [MB] | Accuracy (%) @ 25% Metattack | Accuracy (%) @ 5 nodes nettack |
> |-----------------|-----------------|---------------------|-------------------------|------------------------------|--------------------------------|
> | Pro-GNN         | 1681.17         | 1.01                | 1989                    | 69.72                        | 66.21                          |
> | CSGNN$_{\textrm{noAdj}}$ | 3.29            | 1.32                | 891                     | 70.25                        | 81.90                           |
> | CSGNN           | 9.24            | 5.96                | 1623                    | 74.46                        | 83.29                          |
>
>
> Also, we note that while both our CSGNN and Pro-GNN propose methods to evolve the adjacency matrix, our CSGNN shows significantly lower training computational time, due to our solution taking the form of a learned neural ODE system for the adjacency matrix, while Pro-GNN solves an optimization problem with feedback to the downstream task (which is also an end-to-end solution, although different than ours). For inference, Pro-GNN uses the fixed adjacency matrix found by training, while our CSGNN evolves the input adjacency matrix using the learned network. This can also be seen as an advantage of CSGNN, as it can be used for inference on different kinds of attacks and, therefore, can potentially generalize to different attacks, and this is a future research direction.
>
>
> We also add that investigating the possibility of reducing the computational costs for learning adjacency matrices is an important and interesting topic on its own, that is left for future work.
>
>
> We added the complexity analysis, reported runtimes, and future directions discussion to our revised paper.

---

> ### Author Response · Authors · 2023-11-16
> **Response (Part 4)**
>
> **Regarding attack scope**: We follow the same experimental setting as in [R8], a popular paper whose settings were followed in multiple papers, allowing us to present a broad picture of the adversarial defense capabilities of our CSGNN and compare it with many methods. The Reviewer is correct that our theoretical assumptions are motivated by attacks where the nodes do not change between clean and attacked graphs (i.e., we do not focus on injection attacks). This is also stated in our “Notations” paragraph in Section 3 (“Preliminaries”). Following your question, we revised our Introduction (Section 1) to highlight this fact to enhance the focus of the paper.
> Furthermore, we would like to add that while our focus is on poisoning attacks (i.e., not on injection attacks), in principle, we are unaware of any inherent limitation in our presented theoretical results regarding changing the number of nodes, which translates to injection attacks. Therefore, in principle, one could extend our work to injection attacks. However, as previously discussed, this is not the main focus of our paper, and it is left for future work.
>
> Regarding black box evaluation, we would like to refer the reviewer to our experiments that include random perturbations in Figures 3, 6, and 9.
>
> Regarding the Pubmed dataset, we do not agree that our model achieves suboptimal results. Indeed, our CSGNN does not achieve the highest accuracy on all configurations on Pubmed. Still, a close look at our tables will also suggest that no model is perfect and offers the highest results across all datasets and configurations, and this is also not our claim. We claim that CSGNN offers a powerful model that stems from an ODE perspective and is the first model within the family of ODE-inspired GNNs, at least to our knowledge, to also learn the adjacency matrix. As we discuss in this rebuttal and the paper, this construction is non-trivial as it requires preserving the equivariance and symmetry of the adjacency matrix. Our model indeed offers impressive (also according to most of the reviewers) results on most of the datasets and configurations in this paper. Furthermore, we kindly ask the reviewer to look at Figures 5 and 6, where our CSGNN achieves the highest results on Pubmed, and also in Table 4 our results are among the top 3 models. As previously mentioned, in our experiments, we seek to offer a broad comparison with as many models as possible, and this is the motivation that led to the selection of datasets included in our paper.
>
>
> **Regarding large datasets**: We would like to refer the reviewer to Appendix I (Appendix J in the updated submission), where we provide results on the larger dataset Pubmed. Since our method includes an adaptive mechanism that alters the adjacency matrix in an end-to-end manner, using even larger datasets can be restrictive (memory-wise, due to the $O(n^2)$ possible edges and the need to keep gradients for backpropagation), which is in line with other GNN defense methods that include a learnable adjacency matrix mechanism, see for example [R8]. For reference, we did manage to run our CSGNN on a GPU with 24GB of memory on Pubmed but could not do the same with Pro-GNN [R8] due to out-of-memory errors. This can be attributed to the rather lightweight parameterization of our adjacency matrix update described in Equation (14).
> In addition, we note that employing an adaptive adjacency matrix mechanism to even larger scale datasets is a challenge on its own due to the quadratic complexity in the number of nodes. We agree that it would be interesting to develop methods that can significantly reduce the existing computational costs (which could potentially reduce the complexity of O($n^2$), the maximal number of possible edges). However, as previously discussed in our rebuttal, such development is out of the scope of this paper. The investigation of this research direction can have a large positive impact on the GNN community beyond adversarial defense and is left for future research.
>
> [R1] Stable Architectures for Deep Neural Networks
>
> [R2] Neural Ordinary Differential Equations
>
> [R3] GRAND: Graph Neural Diffusion
>
> [R4] PDE-GCN: Novel Architectures for Graph Neural Networks Motivated by Partial Differential Equations
>
> [R5] GRAND++: Graph Neural Diffusion with A Source Term
>
> [R6] Graph Neural Networks as Gradient Flows: understanding graph convolutions via energy
>
> [R7] Invariant and Equivariant Graph Networks
>
> [R8] Graph Structure Learning for Robust Graph Neural Networks

---

> ### Author Response · Authors · 2023-11-19
> **Reviewer feedback**
>
> Dear Reviewer SPJa,
>
> We sincerely appreciate the time and effort you dedicated to reviewing our paper, as well as the constructive feedback that helped us to improve our revised submission. After providing clarifications to your question, responding to your suggestions, and revising our paper, we hope that the concerns raised by you have been resolved. Could you kindly let us know if our responses have sufficiently addressed your concerns? We are happy to continue the discussion to clarify any standing doubt you may have. We thank you again for your valuable time.
>
> With warm regards,
>
> The authors

---

> > ### Author Response · Authors · 2023-11-21
> > **We would like to have your feedback**
> >
> > Dear Reviewer  SPJa,
> >
> > We would like to once again thank you for the detailed review. During the rebuttal phase we have made significant effort to respond to your queries, and we genuinely hope you found them satisfactory.
> >
> > As the rebuttal phase ends soon, we would like to hear from you, and in case you're indeed satisfied with our responses, if you'd consider to revise your score. Otherwise we are happy to engage in further discussion.
> >
> > Best regards,
> >
> > The authors.

---

> > > ### Comment · Reviewer_SPJa · 2023-12-01
> > > **comments by Reivewer SPJa**
> > >
> > > Thanks for the feedback.  This work still needs to be further improved, and I would like to keep my rating.

---

### Official Review · Reviewer_fwuD · 2023-10-31

**Soundness:** 3 good
**Presentation:** 3 good
**Contribution:** 3 good
**Rating:** 6
**Confidence:** 1

**Summary:**

This paper introduces a novel approach to enhance the robustness of Graph Neural Networks (GNNs) against adversarial perturbations using contractive dynamical systems. The authors establish the mathematical foundations of their architecture, offering theoretical insights into its expected behavior. Through real-world benchmarks, they validate its effectiveness, achieving comparable or superior performance to existing methods. The paper's contributions encompass a new GNN architecture, a theoretical framework for behavior analysis, and empirical proof of its ability to bolster GNN resilience against adversarial attacks.

**Strengths:**

1. The paper addresses the critical issue of improving GNNs' robustness against adversarial attacks. It introduces an innovative approach to enhance Graph Neural Networks' (GNNs) robustness against adversarial perturbations by employing contractive dynamical systems. The simultaneous evolution of node features and adjacency matrices is a unique aspect of this approach, demonstrating a high degree of originality.

2. The paper provides a rigorous mathematical derivation of the proposed architecture and comprehensive empirical evaluations. The authors offer theoretical insights into the expected behavior of their method, and empirical results affirm its effectiveness in bolstering GNNs against adversarial attacks.

3. The paper is well-written, ensuring clarity and accessibility for readers.

**Weaknesses:**

No obvious weaknesses from my perspective.

**Questions:**

1. What are the assumptions behind Theorem 1 & 2?

---

> ### Author Response · Authors · 2023-11-16
> **Response**
>
> We would like to thank the reviewer for the positive feedback, saying that our paper is ‘well-written, ensuring clarity and accessibility for readers’, as well as praising the novelty of our method, stating that ‘The simultaneous evolution of node features and adjacency matrices is a unique aspect of this approach, demonstrating a high degree of originality.’ We now address your question below.  In our revised paper, our responses to your queries are marked in **Red**.
>
>
> **Regarding the assumptions of Theorems 1 and 2:** In our original submission, we state in Appendix A (now turned to Appendix B, in our revised version), after Equation (16) (now Equation (18)), that we assume the activation function $\sigma$ should be monotonically increasing and 1-Lipschitz. We agree that explicitly adding the assumptions to Theorems 1 and 2 improves the quality and clarity of our paper, and we have now added them to our revised version. Please see our changes, marked in Red.

---

> > ### Author Response · Authors · 2023-11-21
> > **We welcome your feedback**
> >
> > Dear Reviewer fwuD,
> >
> > We hope you found our response adequate. We would like to hear your opinion and if you're willing to revise your rating or confidence.
> >
> > Best regards,
> >
> > The authors.

---

> > > ### Comment · Reviewer_fwuD · 2023-12-01
> > >
> > > Thanks for the feedback. I will keep my score.

---

### Official Review · Reviewer_QT3W · 2023-11-01

**Soundness:** 3 good
**Presentation:** 3 good
**Contribution:** 3 good
**Rating:** 6
**Confidence:** 2

**Summary:**

This work introduces massage passing layers in the context of graph representation learning, inspired by differential equations with contractive properties, that have promising capabilities in improving the robustness of GNNs. This claim is then further strengthened by a complete theoretical analysis and extensive benchmark covering many GNN architectures & threat models.

**Strengths:**

- Paper is well written.
- Complete theoretical analysis supported by strong results.

**Weaknesses:**

Unfortunately, the paper is not self-contained for readers with no background in contractive systems and dynamical systems. Since I'm not familiar with these techniques, it's hard for me to point out any weaknesses beyond educated guesses.

**Questions:**

I do not have major questions about the manuscript.

---

> ### Author Response · Authors · 2023-11-16
> **Response**
>
> We thank the reviewer for the overall positive assessment of our submission. The reviewer mentioned the ‘well written’ manner of our paper and the ‘complete theoretical analysis’ with ‘strong results’. Below, we provide our responses to your concerns. In our revised paper, our responses to your queries are marked in **Blue**.
>
>
>
> **Regarding background on contractive systems:** The reviewer is correct that our submission can benefit from an overview of contractive systems. In our original submission, we briefly discussed the meaning of a contractive system in the last paragraph of Section 3 (Preliminaries) and in the Method section (Section 4). Specifically, we provide the discretized definition of a node contractive system in Theorem 2 (please see Equation (10)) and of the adjacency matrix contractive system definition in Equation (11). In our submission, we focus on discretized dynamical systems and therefore only provided those definitions.
>
>
> We also note that Reviewer fwuD highlights the mathematical background provided in our submission:
> “2. The paper provides a rigorous mathematical derivation of the proposed architecture and comprehensive empirical evaluations. The authors offer theoretical insights into the expected behavior of their method, and empirical results affirm its effectiveness in bolstering GNNs against adversarial attacks.”
>
>
> To make our paper more self-contained, following your suggestion, we have added an Appendix section (“Contractive Systems”) that discusses the continuous definition of contractive systems and provides a few of their properties. We now refer to the added Appendix section from our main text in the Preliminaries section. Our changes can be viewed in the updated submission, marked in Blue.

---

> > ### Comment · Reviewer_QT3W · 2023-11-21
> >
> > Thank you for your comments. I've maintained my score.

---

> > > ### Author Response · Authors · 2023-11-21
> > > **Thank you!**
> > >
> > > We appreciate your positive support of our paper.

---

### Author Response · Authors · 2023-11-16
**A note to the Reviewers and Area Chair**

We thank the reviewers for taking the time to read our submission thoroughly.
We appreciate your overall positive feedback, with 3 out of 4 of the reviewers stating our method is **novel** (reviewers fwuD, SPJa, and mRqu), with a **high degree of originality** (reviewer fwuD), and also stating that “The simultaneous evolution of node features and the adjacency matrix is a distinctive feature that can potentially offer intrinsic resistance to adversarial perturbations.” (Reviewer  SPJa).  Also, the reviewers mention the **rigorous theoretical analysis** (reviewers SPJa and fwuD), and **complete theoretical analysis** (reviewer QT3W). Moreover, most reviewers praised our experiments that show **strong results** (reviewer QT3W), and serve as an “empirical proof of its ability to bolster GNN resilience against adversarial attacks” (reviewer fwuD), as well as saying that our method “provide robustness against adversarial attacks.”

Your constructive feedback, insightful suggestions, and questions have enabled us to make changes
to the paper that we believe have improved its overall quality.

We sincerely hope the reviewers will consider raising their scores in light of our changes.

Overall, we have made the following changes, which can be seen in our revised paper:
* We have added an appendix (Appendix A in the updated submission) giving background and context on contractive dynamical systems, as suggested by reviewer QT3W. These additions also address reviewer mRqu’s concerns about the relevance of contractivity to the problem of GNN robustness.
* We have explicitly added assumptions on the activation function for Theorems 1 and 2, answering the question of reviewer fwuD.
* We have included additional references on dynamical systems-based approaches to improve the adversarial robustness of neural networks, following the suggestions of reviewer SPJa. We have included some of these references in the main body of the paper in the introduction and section on related work, but have also added an appendix (Appendix E in the updated submission) where we go into more detail. These references include the reference “On the robustness of graph neural diffusion to topology perturbations”, which reviewer mRqu also mentioned.
* We have added notes in Sections 4.1 and 4.2 to further emphasize the motivation behind our architectural choices to alleviate reviewer SPJa’s concerns about the motivation.
* We have added a comparison of runtime and memory demands of our proposed method compared to Pro-GNN, a representative alternative method, as requested by reviewer SPJa. Additionally, we have added an analysis of the additional computational complexity of our approach versus neural ODE approaches that only evolve the node features. These additions can be found in Appendix K of the revised paper.
* We have added a note further describing why we use the “unit tests” of Mujkanovic et al. (2022) in Section 5.2, addressing a question asked by reviewer mRqu.
* We have added relevant comparisons to GNNGuard’s performance to Appendix J of the revised paper, answering one of reviewer mRqu’s questions.
To provide a focused response, our revised version contains markings of our modifications, using a distinctive color for each reviewer, where applicable.

We are happy to engage in further discussion and are keen to hear your opinion on our response and uploaded revision.

With best regards,

The authors.

---

### Meta-Review · Area_Chair_9YZF · 2023-12-06

**Metareview:**

The authors propose a new defense method for adversarial attacks on GNNs inspired by the contraction systems. Although their analysis are completed and empirical results are better than ProGNN on small graph datasets, the method has two major weaknesses.

1. As raised by reviewers, such a defense method can only work on graph modification attacks. Although authors try to illustrate that their method may also work on such attacks, there is still a lack of evidence for their conclusion. I encourage authors to explore the graph modification attack in their future work since the graph injection attack is more threatening than the graph modification attack.
2. The method is memory-consuming and cannot be implemented on a large graph like OGBN, which was proposed by reviewer SpJa but the author's response is not enough since PubMed is almost 10 times smaller than ogbn-arxiv.

Due to the above weaknesses, I think it's hard to tell whether this method is effective since GNNGuard can also work on graph injection attacks and OGBN-arxiv (and GNNGuard's performance is comparable to the newly proposed method or even better on small graph datasets).

Therefore, I think the method currently should not be accepted as a top-tier conference paper in the current environment.

**Justification For Why Not Higher Score:**

Scenarios are limited.

**Justification For Why Not Lower Score:**

N/A

---

### Decision · Program_Chairs · 2024-01-16

Reject